# Efficient methods for Gaussian Markov random fields under sparse linear constraints

**David Bolin**
King Abdullah University of
Science and Technology
david.bolin@kaust.edu.sa

**Jonas Wallin**
Department of Statistics,
Lund University
jonas.wallin@stat.lu.se

## Abstract

Methods for inference and simulation of linearly constrained Gaussian Markov Random Fields (GMRF) are computationally prohibitive when the number of constraints is large. In some cases, such as for intrinsic GMRFs, they may even be unfeasible. We propose a new class of methods to overcome these challenges in the common case of sparse constraints, where one has a large number of constraints and each only involves a few elements. Our methods rely on a basis transformation into blocks of constrained versus non-constrained subspaces, and we show that the methods greatly outperform existing alternatives in terms of computational cost. By combining the proposed methods with the stochastic partial differential equation approach for Gaussian random fields, we also show how to formulate Gaussian process regression with linear constraints in a GMRF setting to reduce computational cost. This is illustrated in two applications with simulated data.

## 1 Introduction

Linearly constrained Gaussian processes (GPs) have recently gained attention, especially for Gaussian process regression where the model should obey underlying physical principle such as conservation laws or equilibrium conditions [13, 16, 17, 30, 31, 36, 37]. A well-known challenge with these models, and GPs in general, is their high computational cost for inference and prediction in the case of big data sets [2, 9, 14]. One way to reduce computational burden is to impose conditional independence assumptions. In fact, conditional independence between random variables is often explicitly or implicitly assumed in large classes of statistical models including Markov processes, hierarchical models, and graphical models. The assumption typically increases the model's interpretability and facilitates computationally efficient methods for inference [29]. Gaussian variables with conditional independence properties are known as Gaussian Markov random fields (GMRFs), and these are widely used in areas ranging from brain imaging [24, 34, 20] to spatial statistics [7] and time series analysis [28]. GMRFs further arise as computationally efficient approximations of certain GPs [18], which is a fundamental modeling tool in both machine learning and statistics. In particular, such approximations in combination with the integrated nested Laplace approximation (INLA) methodology [29] made latent GMRFs widely used in the applied sciences [3]. GMRFs also have connections with convolutional neural networks leading to recent considerations of deep GMRFs [33].

The key feature of GMRFs that reduces computational cost is sparsity. Specifically, a GMRF

$$\mathbf{X} = [X_1, \ldots, X_n]^\top \sim \mathcal{N}\left(\boldsymbol{\mu}, \mathbf{Q}^{-1}\right), \tag{1}$$

has a sparse precision (inverse covariance) matrix $\mathbf{Q}$ which enables the use of sparse matrix techniques such as sparse Cholesky factorization [8] and iterative methods [35, 23, 34] for computationally efficient sampling and statistical inference [27, 28]. The sparsity is caused by conditional independence assumptions since $Q_{ij} = 0$ if and only if $X_i$ and $X_j$ are independent conditionally on all other variables in $\mathbf{X}$ [see 28, Chapter 2].

35th Conference on Neural Information Processing Systems (NeurIPS 2021).

For a GMRF $\mathbf{X}$, a set of $k$ linear constraints can be formulated as

$$\mathbf{A}\mathbf{X} = \mathbf{b}, \tag{2}$$

where each row in the $k \times n$ matrix $\mathbf{A}$ and the vector $\mathbf{b}$ encodes a constraint on $\mathbf{X}$. Note that observations of GMRFs can be formulated as linear constraints, where, e.g., a point observation of $X_i$ is the simple constraint $X_i = x_i$. These deterministic restrictions are often referred to as hard constraints [28]. If (2) is assumed to hold up to Gaussian noise, i.e., $\mathbf{A}\mathbf{X} \sim \mathcal{N}(\mathbf{b}, \sigma^2 \mathbf{I})$, then the constraints are instead referred to as soft constraints. This scenario is common when GMRFs are incorporated in hierarchical models, where the soft constraints represent noisy observations [28].

Because of the computational advantages of GMRFs, it is of interest to use GMRFs as alternatives to GPs also in the constrained case. Unfortunately, hard constraints can drastically reduce the computational benefits of GMRFs. Specifically, as we will explain in the next section, the current methods for GMRFs with hard constraints in general have a computational cost that scales cubically in the number of constraints $k$. This is prohibitive when there are many constraints, which is common for applications of constrained GPs. For example, physically constrained GPs such as those in [13] have $k = n$ and any GMRF model with $N$ observations without measurement noise will have $k = N$. In the first example the computational cost is cubic in $n$ and in the second it can be cubic in $N$ if the observations are such that $\mathbf{A}$ in (2) is not diagonal, which, e.g., is the case for area observations [21]. In both cases, this renders the computational benefits of the unconstrained GMRF irrelevant. Until now, there has been no methods to circumvent this problem.

**Summary of contributions.** The main contribution of this work is the formulation of a novel class of computationally efficient methods for linearly constrained GMRFs in tasks such as parameter estimation and simulation. These methods explore the conditional independence structure to reduce computational costs compared to traditional methods for situations with large numbers of constraints. The focus is in particular on the case, referred to as sparse hard constraints, when each constraint only involves a few number of variables so that $\mathbf{A}$ is sparse. This is, e.g., common for GPs under physical constraints [13] and for GMRF models with point and area observations [21]. For this case, the main idea is to perform a change of basis so that the constraints are simpler to enforce in the transformed basis. In particular, we show how the change of basis facilitates using the same efficient methods for sparse matrices that are used in the non-constrained case also for the constrained models. In order to use these models for GP regression, the methods are also generalized to GMRF models with both hard and soft constraints. An important feature of the new class of methods, that previous approaches lack, is its applicability to intrinsic GMRFs, which are improper in the sense that a set of eigenvalues of $\mathbf{Q}$ is zero. This makes their distributions invariant to shifts in certain directions, which is a useful property for prior distributions of Bayesian models [28, Chapter 3]. Because of this they are often used in areas such as semiparametric regression [10], medical image analysis [24, 34], and geostatistics [15, 7], and it is thus of interest to be able to enforce linear constraints also for such models. The final contribution is the derivation of GMRFs for constrained GPs, by combining the proposed methods with the stochastic partial differential equation (SPDE) approach by [18] and the nested SPDE methods by [5]. The combined approach is highly computationally efficient compared to standard covariance-based methods, as illustrated in two simulation studies.

**Outline.** In Section 2, the problem is introduced in more detail and the most commonly used methods for sampling and likelihood computations for GMRFs are reviewed. Section 3 introduces the new methods for GMRFs with sparse hard constraints. These methods are extended to the case with both hard and soft constraints in Section 4, followed by the GMRF methods for constrained GPs in Section 5. The methods are illustrated numerically in Section 6. A discussion closes the article, which is supported by a supplementary materials containing proofs and technical details.

## 2   Standard methods for GMRFs under hard constraints

Hard constraints can be divided into interacting and non-interacting constraints. In the latter, (2) specifies a constraint on a subset of the variables in $\mathbf{X}$ so that $\mathbf{A}\mathbf{X} = \mathbf{b}$ can be written as $\mathbf{X}_c = \mathbf{b}_c$, where $c$ denotes a subset of the indices. Specifically, let $\mathbf{X}_u$ denote the remaining variables, then

$$\mathbf{X} = \begin{bmatrix} \mathbf{X}_c \\ \mathbf{X}_u \end{bmatrix} \sim \mathcal{N}\left( \begin{bmatrix} \boldsymbol{\mu}_c \\ \boldsymbol{\mu}_u \end{bmatrix}, \begin{bmatrix} \mathbf{Q}_{cc} & \mathbf{Q}_{cu} \\ \mathbf{Q}_{uc} & \mathbf{Q}_{uu} \end{bmatrix}^{-1} \right) \quad \text{and} \quad \mathbf{X}|\mathbf{A}\mathbf{X} = \mathbf{b} \sim \mathcal{N}\left( \begin{bmatrix} \mathbf{b}_c \\ \boldsymbol{\mu}_{u|c} \end{bmatrix}, \begin{bmatrix} \mathbf{0} & \mathbf{0} \\ \mathbf{0} & \mathbf{Q}_{uu}^{-1} \end{bmatrix} \right),$$

where $\boldsymbol{\mu}_{u|c} = \boldsymbol{\mu}_u - \mathbf{Q}_{uu}^{-1}\mathbf{Q}_{uc}(\mathbf{b_c} - \boldsymbol{\mu}_c)$. Thus, we can split the variables into two subsets and treat the unconditioned variables separately. The more difficult and interesting situation is the case of interacting hard constraints where a simple split of the variables is not possible. From now on, we will assume that we are in this scenario.

Our aim is to construct methods for sampling from the distribution of $\mathbf{X}|\mathbf{AX} = \mathbf{b}$ and for evaluating its log-likelihood function. It is straightforward to show that $\mathbf{X}|\mathbf{AX} = \mathbf{b} \sim \mathcal{N}(\widehat{\boldsymbol{\mu}}, \widehat{\boldsymbol{\Sigma}})$, where $\widehat{\boldsymbol{\mu}} = \boldsymbol{\mu} - \mathbf{Q}^{-1}\mathbf{A}^\top(\mathbf{AQ}^{-1}\mathbf{A}^\top)^{-1}(\mathbf{A}\boldsymbol{\mu} - \mathbf{b})$ and $\widehat{\boldsymbol{\Sigma}} = \mathbf{Q}^{-1} - \mathbf{Q}^{-1}\mathbf{A}^\top(\mathbf{AQ}^{-1}\mathbf{A}^\top)^{-1}\mathbf{AQ}^{-1}$. Since $\widehat{\boldsymbol{\Sigma}}$ has rank $n - k$, likelihood evaluation and sampling is in general expensive. For example, one way is to use an eigenvalue decomposition of $\widehat{\boldsymbol{\Sigma}}$ [see 28, Chapter 2.3.3]. However, this procedure is not practical since it cannot take advantage of the sparsity of $\mathbf{Q}$. Also, for intrinsic GMRFs, $\widehat{\boldsymbol{\Sigma}}$ and $\widehat{\boldsymbol{\mu}}$ cannot be constructed through the expressions above since $\mathbf{Q}^{-1}$ has unbounded eigenvalues.

A commonly used method for sampling under hard linear constraints, sometimes referred to as conditioning by kriging [27], is to first sample $\mathbf{X} \sim \mathcal{N}\left(\boldsymbol{\mu}, \mathbf{Q}^{-1}\right)$ and then correct for the constraints by using $\mathbf{X}^* = \mathbf{X} - \mathbf{Q}^{-1}\mathbf{A}^\top(\mathbf{AQ}^{-1}\mathbf{A}^\top)^{-1}(\mathbf{AX} - \mathbf{b})$ as a sample from the conditional distribution. Here the cost of sampling $\mathbf{X}$ is $\mathcal{C}_\mathbf{Q} + \mathcal{S}_\mathbf{Q}$, where $\mathcal{C}_\mathbf{Q}$ denotes the computational cost of a sparse Cholesky factorization $\mathbf{Q} = \mathbf{R}^\top\mathbf{R}$ and $\mathcal{S}_\mathbf{Q}$ the cost of solving $\mathbf{Rx} = \mathbf{u}$ for $\mathbf{x}$ given $\mathbf{u}$. Adding the cost for the correction step, the total cost of the method is $\mathcal{O}(\mathcal{C}_\mathbf{Q} + (k + 2)\mathcal{S}_\mathbf{Q} + k^3)$.

Let $\pi_{\mathbf{Ax}}(\cdot)$ denote the density of $\mathbf{Ax} \sim \mathcal{N}(\mathbf{A}\boldsymbol{\mu}, \mathbf{AQ}^{-1}\mathbf{A}^\top)$, then the likelihood of $\mathbf{X}|\mathbf{AX} = \mathbf{b}$ can be computed through the expression

$$\pi(\mathbf{x}|\mathbf{Ax} = \mathbf{b}) = \frac{\pi_{\mathbf{Ax}|\mathbf{x}}^*(\mathbf{b}|\mathbf{x})\pi(\mathbf{x})}{\pi_{\mathbf{Ax}}(\mathbf{b})}, \tag{3}$$

where $\pi_{\mathbf{Ax}|\mathbf{x}}^*(\mathbf{b}|\mathbf{x}) = \mathbb{I}(\mathbf{Ax} = \mathbf{b})|\mathbf{AA}^\top|^{-1/2}$ and $\mathbb{I}(\mathbf{Ax} = \mathbf{b})$ denotes the indicator function with $\mathbb{I}(\mathbf{Ax} = \mathbf{b}) = 1$ if $\mathbf{Ax} = \mathbf{b}$ and $\mathbb{I}(\mathbf{Ax} = \mathbf{b}) = 0$ otherwise. This result is formulated in [27] and we provide further details in the supplementary materials by showing that (3) is a density with respect to the Lebesgue measure on the level set $\{\mathbf{x} : \mathbf{Ax} = \mathbf{b}\}$. The computational cost of evaluating the likelihood using this formulation is $\mathcal{O}(\mathcal{C}_\mathbf{Q} + (k + 1)\mathcal{S}_\mathbf{Q} + k^3)$.

Note that these methods only work efficiently for a small number of constraints, because of the term $k^3$ in the computational costs, and for proper GMRFs. In the case of intrinsic GMRFs we cannot work with $\mathbf{AQ}^{-1}\mathbf{A}$ due to the rank deficiency of $\mathbf{Q}$.

## 3 The basis transformation method

In this section, we propose the new class of methods in two steps. We first derive a change of basis in Section 3.1, and then use this to formulate the desired conditional distributions in Section 3.2. The resulting computational costs of likelihood evaluations and simulation are discussed in Section 3.3.

Before stating the results we introduce some basic notation. When working with intrinsic GMRFs the definition in (1) is inconvenient since the covariance matrix has infinite eigenvalues. Instead one can use the canonical parametrization $\mathbf{X} \sim \mathcal{N}_C\left(\boldsymbol{\mu}_C, \mathbf{Q}\right)$, which implies that the density of $\mathbf{X}$ is given by $\pi(\mathbf{x}) \propto \exp\left(-\frac{1}{2}\mathbf{x}^\top\mathbf{Qx} + \boldsymbol{\mu}_C^\top\mathbf{x}\right)$ and thus that $\boldsymbol{\mu}_C = \mathbf{Q}\boldsymbol{\mu}$. Also, since we will be working with non-invertible matrices we will need the Moore–Penrose inverse and the pseudo determinant. We denote the Moore–Penrose inverse of a matrix $\mathbf{B}$ by $\mathbf{B}^\dagger$ and for a symmetric positive semi definite matrix $\mathbf{M}$ we define the pseudo determinant as $|\mathbf{M}|^\dagger = \prod_{i:\lambda_i>0}\lambda_i$ where $\{\lambda_i\}$ are the eigenvalues of $\mathbf{M}$. For the remainder of this article, we will assume the following.

**Assumption 1.** $\mathbf{X} \sim \mathcal{N}_C\left(\mathbf{Q}\boldsymbol{\mu}, \mathbf{Q}\right)$ *where* $\mathbf{Q}$ *is a positive semi-definite* $n \times n$ *matrix with rank* $n - s > 0$ *and null-space* $\mathbf{E}_Q$. $\mathbf{A}$ *is a* $k \times n$ *matrix with rank* $k$ *and we let* $k_0$ *denote* $rank(\mathbf{AE}_Q)$.

Finally, we use the index notation $\mathscr{C} = (1, \ldots, k)$ and $\mathscr{U} = (k + 1, \ldots, n)$ so that, e.g., $\mathbf{v}_\mathscr{C}$ denotes the sub-vector $(v_1, \ldots, v_k)^\top$ of the first $k$ elements in the vector $\mathbf{v}$, $\mathbf{M}_{.,\mathscr{U}}$ ($\mathbf{M}_{\mathscr{U},.}$) denotes the submatrix obtained by extracting the columns (rows) in $\mathbf{M}$ with indices in $\mathscr{U}$, and $\mathbf{M}_{\mathscr{U}\mathscr{U}}$ the submatrix obtained by extracting the columns and rows in $\mathbf{M}$ with indices in $\mathscr{U}$.

## 3.1 Basis construction

Our main idea is to construct a basis on $\mathbb{R}^n$ such that the constraints are easily enforced. A key property of the basis is that $\mathbf{A}$ should be spanned by the first $k$ elements of the basis. Essentially, this means that we are transforming the natural basis into one where the results for the case of non-interacting hard constraints can be used. To construct the basis, note that if $\mathbf{USV}^\top$ is the singular value decomposition (SVD) of a the matrix $\mathbf{A}$, the basis $\mathbf{V}^\top$ is orthonormal and the first $k$ rows span the image of $\mathbf{A}$ and the last $n - k$ rows span the null-space of $\mathbf{A}$. Now, if we let $\mathbf{x}^*$ denote a vector $\mathbf{x}$ expressed in the basis $\mathbf{V}^\top$, then $\mathbf{x}^*$ can be transformed back to the natural basis by $\mathbf{x} = \mathbf{V}\mathbf{x}^*$ hence

$$\mathbf{A}\mathbf{x} = \mathbf{b} \;\;\Leftrightarrow\;\; \mathbf{US}\mathbf{x}^* = \mathbf{b} \;\;\Leftrightarrow\;\; [\mathbf{US}_{\mathscr{C}\mathscr{C}}\;\; \mathbf{0}]\mathbf{x}^* = \mathbf{b} \;\;\Leftrightarrow\;\; \mathbf{US}_{\mathscr{C}\mathscr{C}}\mathbf{x}^*_{\mathscr{C}} = \mathbf{b} \;\;\Leftrightarrow\;\; \mathbf{x}^*_{\mathscr{C}} = \mathbf{b}^*,$$

where $\mathbf{b}^* = (\mathbf{US}_{\mathscr{C}\mathscr{C}})^{-1}\mathbf{b}$. This shows that the SVD is a natural choice of method for constructing the basis of interest, which is defined by an $n \times n$ change-of-basis matrix $\mathbf{T} = \mathbf{V}^\top$.

In Algorithm 1 we present a simple method to build $\mathbf{T}$ based on the SVD. In the algorithm, $id(\mathbf{A})$ is a function that returns the indices of the non-zero columns in $\mathbf{A}$. The computational cost of the method is dominated by that of the SVD, which is $\mathcal{O}\left(k^3 + k^2|id(\mathbf{A})|\right)$ [11, p. 493]. Clearly, the cubic scaling in the number of constraints may reduce the efficiency of any method that requires this basis construction as a first step. However, suppose that the rows of $\mathbf{A}$ can be split into two sub-matrices, $\widetilde{\mathbf{A}}_1$ and $\widetilde{\mathbf{A}}_2$, which have no common non-zero columns. Then the SVD of the two matrices can be computed separately. Suppose now that $\mathbf{A}$ corresponds to $m$ such sub-matrices, then, after reordering, $\mathbf{A} = \left[\widetilde{\mathbf{A}}_1^\top, \ldots, \widetilde{\mathbf{A}}_m^\top\right]^\top$ where $\{\widetilde{\mathbf{A}}_i\}_{i=1}^m$ repre-

---

**Algorithm 1** Constraint basis construction.

**Require:** $\mathbf{A}$ (a $k \times n$ matrix of rank $k$)
1: $\mathbf{T} \leftarrow \mathbf{I}_n$
2: $D \leftarrow id(\mathbf{A})$
3: $\mathbf{USV}^\top \leftarrow svd(\mathbf{A}_{\cdot,D})$
4: $\mathbf{T}_{D,D} \leftarrow \mathbf{V}^\top$
5: $\mathbf{T} \leftarrow [\mathbf{T}_{\cdot,D}\;\; \mathbf{T}_{\cdot,D^c}]$
6: Return $\mathbf{T}$

---

sent sub-constraints such that $id(\widetilde{\mathbf{A}}_i) \cap id(\widetilde{\mathbf{A}}_l) = \emptyset$ for all $i$ and $l$. By replacing the SVD of Algorithm 1 by the $m$ SVDs of the lower-dimensional matrices the computational cost is reduced to $\mathcal{O}\left(\sum_{i=1}^m rank(\widetilde{\mathbf{A}}_i)^3 + rank(\widetilde{\mathbf{A}}_i)^2|id(\widetilde{\mathbf{A}}_i)|\right)$. This method is presented in Algorithm 2. The reordering step is easy to perform and is described in the supplementary materials.

## 3.2 Conditional distributions

Using the change of basis from the previous subsection, we can now derive alternative formulations of the distributions of $\mathbf{AX}$ and $\mathbf{X}|\mathbf{AX} = \mathbf{b}$ which are suitable for sampling and likelihood-evaluation. There are two main results in this section. The first provides an expression of the density of $\mathbf{AX}$ that allows for computationally efficient likelihood evaluations for observations $\mathbf{AX} = \mathbf{b}$. The second formulates the conditional distribution for $\mathbf{X}|\mathbf{AX} = \mathbf{b}$ in a way that allows for efficient sampling of $\mathbf{X}$ given observations $\mathbf{AX} = \mathbf{b}$. To formulate the results, let $\mathbf{T} = CB(\mathbf{A})$ be the output of Algorithm 1 or Algorithm 2 and $\mathbf{X}^* = \mathbf{TX}$ which, under Assumption 1, has distribution $\mathbf{X}^* \sim \mathcal{N}_C\left(\mathbf{Q}^*\boldsymbol{\mu}^*, \mathbf{Q}^*\right)$, where $\boldsymbol{\mu}^* = \mathbf{T}\boldsymbol{\mu}$ and $\mathbf{Q}^* = \mathbf{TQT}^\top$. Henceforth we use stars to denote

---

**Algorithm 2** $CB(\mathbf{A})$. Constraint basis construction for non-overlapping subsets of constraints.

**Require:** $\mathbf{A}$ (a $k \times n$ matrix of rank $k$)
1: $\mathbf{T} \leftarrow \mathbf{I}_n$
2: $D_{full} \leftarrow id(\mathbf{A})$
3: $h \leftarrow 1$
4: $l \leftarrow k + 1$
5: Reorder so that $\mathbf{A} = \left[\widetilde{\mathbf{A}}_1^\top\;\; \ldots\;\; \widetilde{\mathbf{A}}_m^\top\right]^\top$
6: **for** $i = 1 : m$ **do**
7: $\quad D \leftarrow id(\widetilde{\mathbf{A}}_i)$
8: $\quad \mathbf{USV}^\top \leftarrow svd(\widetilde{\mathbf{A}}_i)$
9: $\quad u \leftarrow ncol(\mathbf{U})$
10: $\quad \mathbf{T}_{h:(h+u),D} \leftarrow \left(\mathbf{V}^\top\right)_{1:u,D}$
11: $\quad h \leftarrow h + u + 1$
12: $\quad$ **if** $|D| > u$ **then**
13: $\quad\quad \mathbf{T}_{l:(l+|D|-u),D} \leftarrow \left(\mathbf{V}^\top\right)_{(u+1):|D|,D}$
14: $\quad\quad l \leftarrow l + |D| - u + 1$
15: $\quad$ **end if**
16: **end for**
17: $\mathbf{T} \leftarrow \left[\mathbf{T}_{\cdot,1:|D_{full}|}\;\; (\mathbf{I}_n)_{\cdot,D^c_{full}}\right]$
18: Return $\mathbf{T}$

---

quantities such as means and precisions in the transformed space. Note that we can move from the transformed space to the original space by multiplying with $\mathbf{T}^\top$ for a vector and by multiplying with $\mathbf{T}^\top$ from the left and $\mathbf{T}$ from the right for a matrix.

We use the matrix $\mathbf{H} = \left(\mathbf{A}\mathbf{T}^\top\right)_{\mathscr{C}\mathscr{C}}$, which is equal to $\mathbf{U}\mathbf{S}_{\mathscr{C}\mathscr{C}}$ from the SVD in Algorithm 1, and therefore has inverse $\mathbf{H}^{-1} = \mathbf{S}_{\mathscr{C}\mathscr{C}}^{-1}\mathbf{U}^\top$. Finally, recall that $\mathbf{b}^* = \mathbf{H}^{-1}\mathbf{b}$.

**Theorem 1.** *Under Assumption 1 it follows that*

$$\pi_{\mathbf{A}\mathbf{X}}(\mathbf{b}) = \frac{|\mathbf{Q}^*_{\mathscr{C}|\mathscr{U}}|^{\frac{1}{2}}}{(2\pi)^{k/2}|\mathbf{A}\mathbf{A}^\top|^{1/2}} \cdot \exp\left(-\frac{1}{2}\left(\mathbf{b}^* - \boldsymbol{\mu}^*_{\mathscr{C}}\right)^\top \mathbf{Q}^*_{\mathscr{C}|\mathscr{U}}\left(\mathbf{b}^* - \boldsymbol{\mu}^*_{\mathscr{C}}\right)\right),$$

*where $\mathbf{Q}^*_{\mathscr{C}|\mathscr{U}} = \mathbf{Q}^*_{\mathscr{C}\mathscr{C}} - \mathbf{Q}^*_{\mathscr{C}\mathscr{U}}\left(\mathbf{Q}^*_{\mathscr{U}\mathscr{U}}\right)^\dagger \mathbf{Q}^*_{\mathscr{U}\mathscr{C}}$ and $|\mathbf{Q}^*_{\mathscr{C}|\mathscr{U}}|^{\frac{1}{2}} = |\mathbf{Q}|^{\frac{1}{2}}|\mathbf{Q}^*_{\mathscr{U}\mathscr{U}}|^{-\frac{1}{2}}$.*

If $\mathbf{Q}$ is positive definite we have $\mathbf{Q}^*_{\mathscr{C}|\mathscr{U}} = \mathbf{Q}^*_{\mathscr{C}\mathscr{C}} - \mathbf{Q}^*_{\mathscr{C}\mathscr{U}}\left(\mathbf{Q}^*_{\mathscr{U}\mathscr{U}}\right)^{-1}\mathbf{Q}^*_{\mathscr{U}\mathscr{C}}$ and we can then replace $|\mathbf{Q}^*_{\mathscr{C}|\mathscr{U}}|^{\frac{1}{2}}$ with $|\mathbf{Q}^*_{\mathscr{C}|\mathscr{U}}|^{\frac{1}{2}}$ in the expression of $\pi_{\mathbf{A}\mathbf{X}}(\mathbf{b})$.

**Theorem 2.** *Under Assumption 1 it follows that*

$$\mathbf{X}|\mathbf{A}\mathbf{X} = \mathbf{b} \sim \mathcal{N}_C\left(\mathbf{Q}_{X|b}\widetilde{\boldsymbol{\mu}}, \mathbf{Q}_{X|b}\right)\mathbb{I}\left(\mathbf{A}\mathbf{X} = \mathbf{b}\right), \tag{4}$$

*where $\mathbf{Q}_{X|b} = \mathbf{T}_{\mathscr{U}}^\top\mathbf{Q}^*_{\mathscr{U}\mathscr{U}}\mathbf{T}_{\mathscr{U}}$, is positive semi-definite with rank $n - s - (k - k_0)$ and $\widetilde{\boldsymbol{\mu}} = \mathbf{T}^\top\widetilde{\boldsymbol{\mu}}^*$ with $\widetilde{\boldsymbol{\mu}}^* = \left[\begin{smallmatrix}\mathbf{b}^*\\ \boldsymbol{\mu}^*_{\mathscr{U}} - \mathbf{Q}^{*\dagger}_{\mathscr{U}\mathscr{U}}\mathbf{Q}^*_{\mathscr{U}\mathscr{C}}(\mathbf{b}^* - \boldsymbol{\mu}^*_{\mathscr{C}})\end{smallmatrix}\right]$.*

Note that $\mathbf{Q}_{X|b}\mathbf{A} = \mathbf{0}$, which implies that the right side of (4) is a (possibly intrinsic) density with respect to Lebesgue measure on the level set $\{\mathbf{x} : \mathbf{A}\mathbf{x} = \mathbf{b}\}$. Further, note that $\mathbf{Q}^*_{\mathscr{U}\mathscr{U}}\mathbf{T}_{\mathscr{U}}\mathbf{E}_0 = \mathbf{0}$, which implies that $\mathbf{X}$ is improper on the span of $\mathbf{T}_{\mathscr{U}}\mathbf{E}_0$.

If $\mathbf{Q}$ is positive definite we get the following corollary.

**Corollary 1.** *Under Assumption 1 with $s = 0$, we have*

$$\mathbf{X}|\mathbf{A}\mathbf{X} = \mathbf{b} \sim \mathcal{N}\left(\widetilde{\boldsymbol{\mu}}, \widetilde{\boldsymbol{\Sigma}}\right)\mathbb{I}\left(\mathbf{A}\mathbf{X} = \mathbf{b}\right), \tag{5}$$

*where $\widetilde{\boldsymbol{\Sigma}} = \mathbf{T}_{\mathscr{U}}^\top\left(\mathbf{Q}^*_{\mathscr{U}\mathscr{U}}\right)^{-1}\mathbf{T}_{\mathscr{U}}$, is a positive semi-definite matrix of rank $n - k$ and $\widetilde{\boldsymbol{\mu}} = \mathbf{T}^\top\widetilde{\boldsymbol{\mu}}^*$ with $\widetilde{\boldsymbol{\mu}}^* = \left[\begin{smallmatrix}\mathbf{b}^*\\ \boldsymbol{\mu}^*_{\mathscr{U}} - (\mathbf{Q}^*_{\mathscr{U}\mathscr{U}})^{-1}\mathbf{Q}^*_{\mathscr{U}\mathscr{C}}(\mathbf{b}^* - \boldsymbol{\mu}^*_{\mathscr{C}})\end{smallmatrix}\right]$.*

### 3.3 Sampling and likelihood evaluations

The standard method for sampling a GMRF $\mathbf{X} \sim \mathcal{N}\left(\boldsymbol{\mu}, \mathbf{Q}^{-1}\right)$ is to first compute the Cholesky factor $\mathbf{R}$ of $\mathbf{Q}$, then sample $\mathbf{Z} \sim \mathcal{N}\left(\mathbf{0}, \mathbf{I}\right)$, and finally set

$$\mathbf{X} = \boldsymbol{\mu} + \mathbf{R}^{-1}\mathbf{Z}. \tag{6}$$

To sample $\mathbf{X}|\mathbf{A}\mathbf{X} = \mathbf{b}$ we use this method in combination with Theorem 2 as shown in Algorithm 3. The cost of using the algorithm for sampling, and for computing the expectation of $\mathbf{X}$ in Theorem 2, is dominated by $\mathcal{C}_{\mathbf{Q}^*_{\mathscr{U}\mathscr{U}}}$ given that $\mathbf{T}$ has been pre-computed. Similarly, the cost for evaluating the likelihood in Theorem 1 is dominated by the costs of the Cholesky factors $\mathcal{C}_{\mathbf{Q}^*_{\mathscr{U}\mathscr{U}}} + \mathcal{C}_{\mathbf{Q}} + \mathcal{C}_{\mathbf{A}\mathbf{A}^\top}$.

These costs are not directly comparable to costs of the methods from Section 2 since they involve operations with the transformed precision matrix $\mathbf{Q}^* = \mathbf{T}\mathbf{Q}\mathbf{T}^\top$ which may have a different, and often denser, sparsity structure than $\mathbf{Q}$. In fact if $\mathbf{T}$ is dense the method will not be practically useful since even the construction of $\mathbf{Q}^*$ would be $\mathcal{O}\left(n^2\right)$. Thus, to understand the computational cost we must understand the sparsity structure of the transformed matrix. To that end, first note that only the rows $id(\mathbf{A})$ in $\mathbf{Q}^*$ will have a sparsity structure that is different from that in $\mathbf{Q}$. In general, the variables involved for the $i$th constraint, $id(\mathbf{A}_i\mathbf{X})$, will in the constrained distribution share all their neighbors. This implies that if $i \in id(\mathbf{A})$, then $|\mathbf{Q}^*_{i,j}| > 0$ if $|\mathbf{Q}_{i,j}| > 0$ and we might have $|\mathbf{Q}^*_{i,j}| > 0$ if $\sum_{k\in id(\mathbf{A})}|\mathbf{Q}^*_{k,j}| > 0$. This provides a worst-case scenario for the amount of non-zero elements in $\mathbf{Q}^*$, where we see that the sparsity of the constraints is important.

# 4 GMRFs under hard and soft constraints

As previously mentioned, one can view observations of a GMRF as hard constraints. In many cases, these observations are assumed to be taken under Gaussian measurement noise, which can be seen as soft constraints on the GMRF. It is therefore common to have models with both soft and hard constraints (e.g., a model with noisy observations of a field with local sum-to-zero constraints). Here, we extend the methods of the previous section to this case. Specifically, we consider the following hierarchical model

$$\mathbf{X} \sim \mathcal{N}_C\left(\mathbf{Q}\boldsymbol{\mu}, \mathbf{Q}\right), \quad \text{subject to } \mathbf{AX} = \mathbf{b},$$
$$\mathbf{Y} \sim \mathcal{N}\left(\mathbf{BX}, \sigma_Y^2 \mathbf{I}\right), \tag{7}$$

where $\mathbf{Y} \in \mathbb{R}^m$ represent noisy observations of the linear combinations $\mathbf{BX}$ of $\mathbf{X} \in \mathbb{R}^n$, with $m \le n$, satisfying Assumption 1, and $\mathbf{B}$ is an $m \times n$ matrix with rank $m$. To deal with this type of models we present two results in this section. First, Theorem 3 shows how to compute the likelihood of the model. Second, the result in Theorem 4 can be used to efficiently compute the mean of $\mathbf{X}$ given the constraints and to sample from it.

We use the hat notation – like $\widehat{\mathbf{Q}}$ – to denote quantities for distributions conditionally on the observations $\mathbf{Y} = \mathbf{y}$. We also use the notation from Theorem 2 and additionally introduce $\mathbf{B}^* = \mathbf{B}\mathbf{T}^\top$ and $\mathbf{y}^* = \mathbf{y} - \mathbf{B}\mathbf{T}_{\mathscr{C}}^\top \mathbf{b}^*$. We start by deriving the likelihood, $\pi_{\mathbf{Y}|\mathbf{AX}}(\mathbf{y}|\mathbf{b})$, which is needed for inference.

**Theorem 3.** *For the model in* (7) *one has*

$$\pi_{\mathbf{Y}|\mathbf{AX}}(\mathbf{y}|\mathbf{b}) = \frac{\sigma_Y^{-m}|\mathbf{Q}_{\mathscr{U}\mathscr{U}}^*|^{\frac{\dagger}{2}}}{(2\pi)^{c_0}\,|\widehat{\mathbf{Q}}_{\mathscr{U}\mathscr{U}}^*|^{\frac{\dagger}{2}}} \exp\left(-\frac{1}{2}\left[\frac{\mathbf{y}^{*T}\mathbf{y}^*}{\sigma_Y^2} + \widetilde{\boldsymbol{\mu}}_{\mathscr{U}}^{*\top}\mathbf{Q}_{\mathscr{U}\mathscr{U}}^*\widetilde{\boldsymbol{\mu}}_{\mathscr{U}}^* - \widehat{\boldsymbol{\mu}}_{\mathscr{U}}^{*\top}\widehat{\mathbf{Q}}_{\mathscr{U}\mathscr{U}}^*\widehat{\boldsymbol{\mu}}_{\mathscr{U}}^*\right]\right),$$

*where $c_0 > 0$, and*

$$\widehat{\mathbf{Q}}_{\mathscr{U}\mathscr{U}}^* = \mathbf{Q}_{\mathscr{U}\mathscr{U}}^* + \frac{1}{\sigma_Y^2}\left(\mathbf{B}_{\mathscr{U}}^*\right)^\top \mathbf{B}_{\mathscr{U}}^*,$$
$$\widehat{\boldsymbol{\mu}}_{\mathscr{U}}^* = \widehat{\mathbf{Q}}_{\mathscr{U}\mathscr{U}}^{*\dagger}\left(\mathbf{Q}_{\mathscr{U}\mathscr{U}}^*\widetilde{\boldsymbol{\mu}}_{\mathscr{U}}^* + \frac{1}{\sigma_Y^2}\left(\mathbf{B}_{\mathscr{U}}^*\right)^\top \mathbf{y}^*\right). \tag{8}$$

The computational cost of evaluating the likelihood is $\mathcal{C}_{\widehat{\mathbf{Q}}_{\mathscr{U}\mathscr{U}}^*} + \mathcal{S}_{\widehat{\mathbf{Q}}_{\mathscr{U}\mathscr{U}}^*} + \mathcal{C}_{\mathbf{Q}_{\mathscr{U}\mathscr{U}}^*}$. The following theorem introduces the distribution of $\mathbf{X}$ given the event $\{\mathbf{AX} = \mathbf{b}, \mathbf{Y} = \mathbf{y}\}$, which for example is needed when the model is used for prediction.

**Theorem 4.** *For $\mathbf{X}$ in* (7) *one has* $\mathbf{X}|\{\mathbf{AX} = \mathbf{b}, \mathbf{Y} = \mathbf{y}\} \sim \mathcal{N}\left(\widehat{\boldsymbol{\mu}}, \widehat{\mathbf{Q}}\right)$ *where* $\widehat{\mathbf{Q}} = \mathbf{T}_{\mathscr{U}}^\top \widehat{\mathbf{Q}}_{\mathscr{U}\mathscr{U}}^* \mathbf{T}_{\mathscr{U}}$, *and* $\widehat{\boldsymbol{\mu}} = \mathbf{T}^\top \begin{bmatrix} \mathbf{b}^* \\ \widehat{\boldsymbol{\mu}}_{\mathscr{U}}^* \end{bmatrix}$. *Here $\widehat{\mathbf{Q}}_{\mathscr{U}\mathscr{U}}^*$ and $\widehat{\boldsymbol{\mu}}_{\mathscr{U}}^*$ are given in* (8). *Further, let $\mathbf{E}_{\mathbf{Q}_{\mathscr{U}\mathscr{U}}^*}$ be the null space of $\mathbf{Q}_{\mathscr{U}\mathscr{U}}^*$ then $\mathrm{rank}(\widehat{\mathbf{Q}}) = n - s - (k - k_0) + rank(\mathbf{B}_{\mathscr{U}}^*\mathbf{E}_{\mathbf{Q}_{\mathscr{U}\mathscr{U}}^*})$.*

Since the distribution in the theorem is a normal distribution, we can sample from $\mathbf{X}$ given the event $\{\mathbf{AX} = \mathbf{b}, \mathbf{Y} = \mathbf{y}\}$ using sparse Cholesky factorization as shown in Algorithm 4.

# 5 Constrained Gaussian processes and the SPDE approach

Gaussian processes and random fields are typically specified in terms of their mean and covariance functions. However, a problem with any covariance-based Gaussian model is the computational cost for inference and simulation. Several authors have proposed solutions to this problem, and one particularly important solution is the GMRF approximation by [18]. This method is applicable to GPs and random fields with Matérn covariance functions,

$$r(h) = \frac{\sigma^2}{\Gamma(\nu)2^{\nu-1}}(\kappa h)^\nu K_\nu(\kappa h), \qquad h \ge 0,$$

which is the most popular covariance model is spatial statistics, inverse problems and machine learning [12, 26]. The method relies on the fact that a Gaussian random field $X(s)$ on $\mathbb{R}^d$ with a Matérn covariance function can be represented as a solution to the SPDE

$$(\kappa^2 - \Delta)^{\frac{\alpha}{2}} X = \phi \mathcal{W}, \tag{9}$$

| **Algorithm 3** Sampling $\mathbf{X} \sim \mathcal{N}\left(\boldsymbol{\mu}, \mathbf{Q}^{-1}\right)$ subject to $\mathbf{AX} = \mathbf{b}$. | **Algorithm 4** Sampling $\mathbf{X} \sim \mathcal{N}\left(\boldsymbol{\mu}, \mathbf{Q}^{-1}\right)$ subject to $\mathbf{AX} = \mathbf{b}$ and $\mathbf{Y} = \mathbf{y}$ where $\mathbf{Y} \sim \mathcal{N}\left(\mathbf{BX}, \sigma_Y^2 \mathbf{I}\right)$. |
|---|---|
| **Require:** $\mathbf{A}, \mathbf{b}, \mathbf{Q}, \boldsymbol{\mu}, \mathbf{T}$ | **Require:** $\mathbf{A}, \mathbf{b}, \mathbf{Q}, \boldsymbol{\mu}, \mathbf{T}, \mathbf{y}, \mathbf{B}, \sigma_Y^2$ |

**Algorithm 3**

1: $\mathscr{C} \leftarrow 1 : nrow(\mathbf{A})$
2: $\mathscr{U} \leftarrow (nrow(\mathbf{A}) + 1) : ncol(\mathbf{A})$
3: $\mathbf{Q}^* \leftarrow \mathbf{TQT}^\top$
4: $\mathbf{R} \leftarrow chol(\mathbf{Q}^*_{\mathscr{U}\mathscr{U}})$
5: $\mathbf{b}^* \leftarrow solve\left(\left(\mathbf{AT}^\top\right)_{\mathscr{C}\mathscr{C}}, \mathbf{b}\right)$
6: $\mathbf{m}^* \leftarrow \boldsymbol{\mu}_{\mathscr{U}} - solve(\mathbf{R}^\top, \mathbf{Q}^*_{\mathscr{U}\mathscr{C}}(\mathbf{b}^* - \mathbf{T}_{\mathscr{C}}\boldsymbol{\mu}))$
7: Sample $\mathbf{Z} \sim \mathcal{N}\left(\mathbf{0}, \mathbf{I}_{\mathscr{U}\mathscr{U}}\right)$
8: $\mathbf{X}^* \leftarrow [\mathbf{b}^*, solve(\mathbf{R}, \mathbf{m}^* + \mathbf{Z})]^\top$
9: $\mathbf{X} \leftarrow \mathbf{T}^\top \mathbf{X}^*$
10: Return $\mathbf{X}$

**Algorithm 4**

1: $\mathscr{C} \leftarrow 1 : nrow(\mathbf{A})$
2: $\mathscr{U} \leftarrow (nrow(\mathbf{A}) + 1) : ncol(\mathbf{A})$
3: $\mathbf{Q}^* \leftarrow \mathbf{TQT}^\top$
4: $\mathbf{B}^* \leftarrow \mathbf{BT}^\top$
5: $\mathbf{R} \leftarrow chol(\mathbf{Q}^*_{\mathscr{U}\mathscr{U}} + \frac{1}{\sigma_Y^2}(\mathbf{B}^*)^\top \mathbf{B}^*)$
6: $\mathbf{b}^* \leftarrow solve\left(\left(\mathbf{AT}^\top\right)_{\mathscr{C}\mathscr{C}}, \mathbf{b}\right)$
7: $\mathbf{y}^* \leftarrow \mathbf{y} - \mathbf{BT}_{\mathscr{C}}^\top \mathbf{b}^*$
8: $\mathbf{m}^* \leftarrow solve\left(\mathbf{R}^\top, \mathbf{Q}^*_{\mathscr{U}\mathscr{U}}\mathbf{T}_{\mathscr{U}}\boldsymbol{\mu} + \frac{1}{\sigma_Y^2}(\mathbf{B}^*)^\top \mathbf{y}^* - \mathbf{Q}^*_{\mathscr{U}\mathscr{C}}(\mathbf{b}^* - \mathbf{T}_{\mathscr{C}}\boldsymbol{\mu})\right)$
9: Sample $\mathbf{Z} \sim \mathcal{N}\left(\mathbf{0}, \mathbf{I}_{\mathscr{U}\mathscr{U}}\right)$
10: $\mathbf{X}^* \leftarrow [\mathbf{b}^*, solve(\mathbf{R}, \mathbf{m}^* + \mathbf{Z})]^\top$
11: $\mathbf{X} \leftarrow \mathbf{T}^\top \mathbf{X}^*$
12: Return $\mathbf{X}$

where the exponent $\alpha$ is related to $\nu$ via the relation $\alpha = \nu + d/2$, $\Delta$ is the Laplacian, $\mathcal{W}$ is Gaussian white noise on $\mathbb{R}^d$, and $\phi$ is a constant that controls the variance of $X$. The GMRF approximation by [18] is based on restricting (9) to a bounded domain $\mathcal{D}$, imposing homogeneous Neumann boundary conditions on the operator, and approximating the solution via a finite element method (FEM). The resulting approximation is $X_h(s) = \sum_{i=1}^n X_i \varphi_i(s)$, where $\{\varphi_i(s)\}$ are piecewise linear basis functions induced by a triangulation of the domain, and the vector $\mathbf{X}$ with all weights $X_i$ is a centered multivariate Gaussian distribution. This can be done for any $\alpha > d/2$ [4], but the case $\alpha \in \mathbb{N}$ is of particular importance since $\mathbf{X}$ then is a GMRF. In particular, when $\alpha = 2$ and $\phi = 1$, the precision matrix of $\mathbf{X}$ is $\mathbf{Q} = (\kappa^2 \mathbf{C} + \mathbf{G})\mathbf{C}^{-1}(\kappa^2 \mathbf{C} + \mathbf{G})$, where $\mathbf{C}$ is a diagonal matrix with diagonal elements $C_{ii} = \int \varphi_i(s)ds$, and $\mathbf{G}$ is a sparse matrix with elements $G_{ij} = \int \varphi_i(s)\varphi_j(s)ds$.

Clearly, a linear constraint on $X_h(s)$ can be written as a constraint on $\mathbf{X}$. For example, if $X_h(s)$ is observed at a location in a given triangle, it creates a linear constraint on the three variables in $\mathbf{X}$ corresponding to the corners of the triangle. Thus, if we draw some observation locations $s_1, \ldots, s_k$ in the domain, we can write $\mathbf{Y} = (X_h(s_1), \ldots, X_h(s_k))^\top = \mathbf{A}_Y \mathbf{X}$ where $\mathbf{A}_Y$ is a $k \times n$ matrix with $(\mathbf{A}_Y)_{ij} = \varphi_j(s_i)$. A model where a Gaussian Matérn fields is observed without measurement noise can therefore be handled efficiently by combining the SPDE approach with the methods from Section 3. The next section contains a simulation study that compares this combined approach with a standard covariance-based approach in terms of computational cost.

Through the nested SPDE approach in [5] one can also construct computationally efficient representations of differentiated Gaussian Matérn fields like $U(\mathbf{s}) = (\mathbf{v}^\top \nabla)X(\mathbf{s})$, where $\mathbf{v}^\top \nabla$ is the directional derivative in the direction given by the vector $\mathbf{v}$ and $X(\mathbf{s})$ is a sufficiently differentiable Matérn field. A FEM approximation of this model can be written as $U_h(s) = \sum_{i=1}^n U_i \varphi_i(s)$ where now $\mathbf{U} \sim \mathcal{N}(\mathbf{0}, \mathbf{A}_U \mathbf{Q}^{-1} \mathbf{A}_U^\top)$. Here $\mathbf{A}_U$ is a sparse matrix representing the directional derivative and $\mathbf{Q}$ is the precision matrix of the GMRF representation of $X(\mathbf{s})$ [5]. If we introduce $\mathbf{X} \sim \mathcal{N}(\mathbf{0}, \mathbf{Q}^{-1})$, we may write $\mathbf{U} = \mathbf{A}_U \mathbf{X}$, and we can thus enforce a restriction on the directional derivative of $\mathbf{X}$ as a linear restriction $\mathbf{A}_U \mathbf{X} = \mathbf{b}$. As an example, $\mathbf{v} = (1, 1)^\top$ and $\mathbf{b} = \mathbf{0}$ results in the restriction $\frac{\partial}{\partial s_1}X(\mathbf{s}) + \frac{\partial}{\partial s_2}X(\mathbf{s}) = 0$, or in other words that the field is divergence-free. In the next section we use this in combination with the methods in Section 4 to construct a computationally efficient Gaussian process regression under linear constraints.

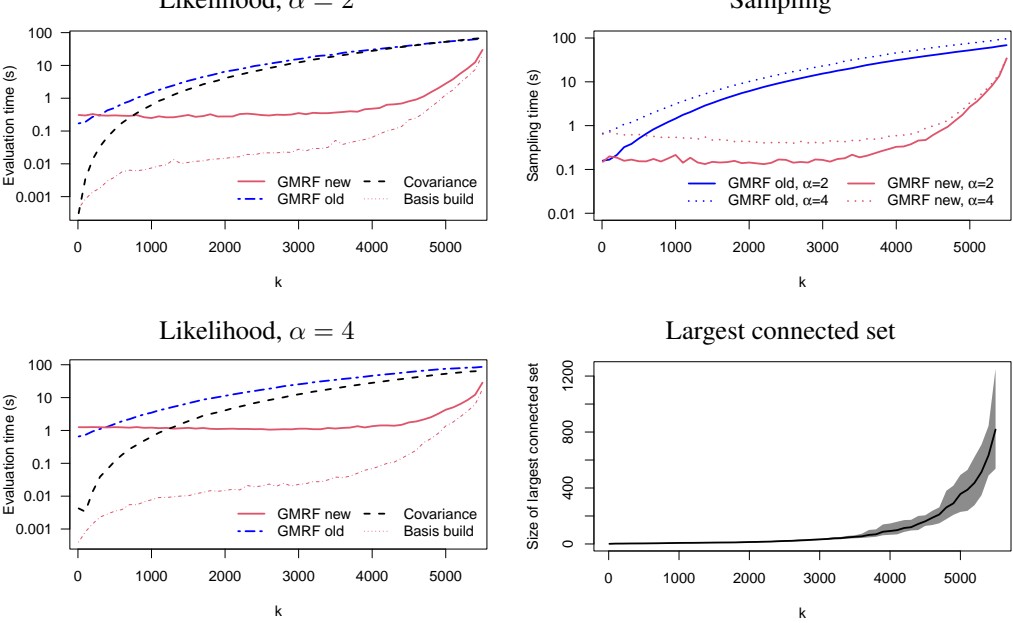

Figure 1: Average computation times (based on 10 replications) for one likelihood evaluation (left) and one sample from $\mathbf{X}|\mathbf{A}_Y\mathbf{X} = \mathbf{y}$ (top right) of the Matérn model with $\alpha = 2$ and $\alpha = 4$ as functions of the number of observations $k$. The computation times for the new GMRF method includes the time needed to construct the basis matrix $\mathbf{T}$, which also is shown separately and which depends on the largest set of connected observations (bottom right).

## 6 Numerical illustrations

In this section we present two applications. In both cases, timings are obtained through R [25] implementations, available in the CB R package [6], run on an iMac Pro computer with a 3.2 GHz Intel Xeon processor.

### 6.1 Observations as hard constraints

Suppose that we have an SPDE approximation $X_h(s)$ of a Gaussian Matérn field $X(s)$, as described above with $\mathcal{D} = [0, 1]^2$ and a triangulation for the GMRF approximation that is based on a uniform mesh with $100 \times 100$ nodes in $\mathcal{D}$. We consider the costs of sampling $X_h(s)$ conditionally on $k$ point observations without measurement noise, and of log-likelihood evaluations for these observations. In both cases, the observations are simulated using the parameters $\kappa^2 = 0.5, \phi = 1$ and $\alpha = 2$ or $\alpha = 4$.

In the top left panel of Figure 1 we show the computation time for one log-likelihood evaluation using the standard method from Section 2 on the GMRF $\mathbf{X}$ of weights for the basis expansion of $X_h(\mathbf{s})$. The panel also shows the corresponding computation time for the new method from Section 3. The computation times are evaluated for different values of $k$, where for each $k$ the observation locations are sampled uniformly over triangles, and uniformly within triangles, under the restriction that there can be only one observation per triangle, which guarantees that $\mathbf{A}_Y\mathbf{Q}^{-1}\mathbf{A}_Y^\top$ has full rank. In each iteration, the values of $\kappa^2$ and $\phi$ that are evaluated in the likelihood are sampled from a uniform distribution on $[1, 2]$. The curves shown in the figure are computed as averages of 10 repetitions for each value of $k$. As a benchmark, we show the time it takes to evaluate the log-likelihood assuming that $X(s)$ is a Gaussian Matérn field, which means that we evaluate the log-likelihood $\ell(\mathbf{Y})$ of a $k$-dimensional $\mathcal{N}(\mathbf{0}, \boldsymbol{\Sigma})$ distribution without using any sparsity properties. This is done by calculating the Cholesky factor $\mathbf{R}$ of $\boldsymbol{\Sigma}$ and then computing $\ell(\mathbf{Y}) = -\sum_{i=1}^{k} \log R_{ii} - \frac{1}{2}\mathbf{Y}^\top\mathbf{R}^{-1}\mathbf{R}^{-\top}\mathbf{Y}$.

An alternative method for this example would be to adapt the mesh to the observation locations, so that one only has observations at mesh nodes. This is, however, in general not a good solution as it might require using poor meshes (from a numerical stability point of view) or using meshes with a large number of nodes, increasing the computational cost, and hence we do not explore it here.

The covariance-based method is the fastest up to approximately $k = 1000$ observations, since the problem then is too small for sparsity to be beneficial, whereas the new method is fastest for $k > 1000$. If $k$ is large relative to the total size of the GMRF, constraints may no longer be local since many constraints will be interacting. As illustrated in the bottom right panel of Figure 1, this causes the largest set of connected observations to increase rapidly for $k > 4000$. The computation time for creating $\mathbf{T}$ will roughly scale cubically in the size of the largest set of connected observations, which is the main reason for the large increase in the computation time for the new method when $k > 5000$. However, it should be noted that the construction of the basis needed for the new method only has to be done once, so even though the computation time for the construction of $\mathbf{T}$ is high when the largest set of connected observations becomes large, the method will still be efficient for maximum likelihood estimation using numerical optimization, where several likelihood evaluations are needed.

The specific values of $\kappa$ and $\phi$ have no effect on the computational costs for any of the methods. The only parameter that affects the results is $\alpha$ since it changes the sparsity structure of the GMRF where a larger value results in precision matrices with more non-zero elements. The bottom left panel in Figure 1 shows the results for $\alpha = 4$, where we can note that the standard GMRF method is slower than the covariance-based method for all values of $k$, whereas our proposed method still performs the best for large values of $k$. In the supplementary materials, corresponding results for $\alpha = 1$ and $\alpha = 3$ are shown with similar results.

In the top right panel of Figure 1 we show the time needed to sample $X_h(\mathbf{s})$ conditionally on the observations $\mathbf{A}_Y \mathbf{X} = \mathbf{y}$, i.e., to simulate from $\mathbf{X} | \mathbf{A}_Y \mathbf{X} = \mathbf{y}$. Both the old method (conditioning by kriging) and the new method (using (6) with mean and precision from Theorem 2) are shown. We do not show the covariance-based method since it is much slower than both GMRF methods. Also here the displayed values are averages of 10 repetitions for each value of $k$, and for each repetition the simulation is performed using values of $\kappa^2$ and $\phi$ that sampled from a uniform distribution on $[1, 2]$. The results are again shown for $\alpha = 2$ and $\alpha = 4$, and the results for $\alpha = 1, 3$ are presented in the supplementary materials. The results are similar to those for likelihood evaluations, where the new method is much faster for large numbers of observations.

## 6.2   Gaussian process regression with linear constraints

We now consider an application to GP regression from [13], in which one is given noisy observations $\mathbf{Y}_i = \mathbf{f}(\mathbf{s}_i) + \boldsymbol{\varepsilon}_i$, with $\boldsymbol{\varepsilon}_i \sim \mathcal{N}(0, \sigma_e^2 \mathbf{I})$ of a bivariate function $\mathbf{f} = (f_1, f_2) : \mathbb{R}^2 \to \mathbb{R}^2$ where $f_1(\mathbf{s}) = e^{-as_1 s_2}(as_1 \sin(s_1 s_2) - s_1 \cos(s_1 s_2))$ and $f_2(\mathbf{s}) = e^{-as_1 s_2}(s_2 \sin(s_1 s_2) - as_2 \sin(s_1 s_2))$. The goal is to use Gaussian process regression to reconstruct $\mathbf{f}$, under the assumption that we know that it is divergence-free, i.e., $\frac{\partial}{\partial s_1} \mathbf{f} + \frac{\partial}{\partial s_2} \mathbf{f} = 0$. We thus want to improve the regression estimate by incorporating this information in the GP prior for $\mathbf{f}$. This can be done as in [32, 37, 13] by encoding the information directly in the covariance function, or by imposing the restriction through a hard constraint at each spatial location through the nested SPDE approach. This is done by setting $\mathbf{B} = \mathbf{A}_Y$ and $\mathbf{A} = \mathbf{A}_U$ in (7) where the matrices $\mathbf{A}_Y$ and $\mathbf{A}_U$ are defined in Section 5.

Following [13], we choose $a = 0.01$ and $\sigma_2 = 10^{-4}$ and generate 50 observations at randomly selected locations in $[0, 4] \times [0, 4]$ and predict the function $\mathbf{f}$ at $N^2 = 20^2$ regularly spaced locations in the square. Independent Matérn priors with $\alpha = 4$ are assumed for $f_1$ and $f_2$ and the covariance-based approach by [13] is taken as a baseline method. As an alternative, we consider the SPDE approximation of the Matérn priors, with $n$ basis functions obtained from a regular triangulation of an extended domain $[-2, 6] \times [-2, 6]$ (the extension is added to reduce boundary effects). To be able to use Algorithm 2, we only enforce the divergence constraint at every third node for the SPDE model. This procedure can be seen as an approximation of the divergence operator that will converge to the true operator when the number of basis functions increases.

The parameters of the baseline model and of the SPDE model are estimated using maximum likelihood, where the likelihood for the SPDE model is computed using Theorem 3. The function $\mathbf{f}$ is then reconstructed using the posterior mean of the GP given the data, which is calculated using Theorem 4. This experiment is repeated for 50 randomly generated datasets. In the left panel of Figure 2 we show the average root mean squared error (RMSE) for the reconstruction of $\mathbf{f}$ for the SPDE model as a function of $n$, based on these 50 simulations, together with the corresponding RMSE of the baseline method. The shaded region for the SPDE model is a pointwise $95\%$ confidence band. One can see that the SPDE model gives a comparable RMSE as long as $n$ is large enough.

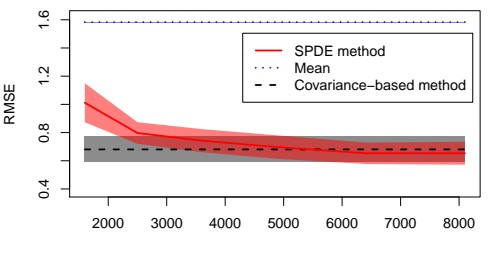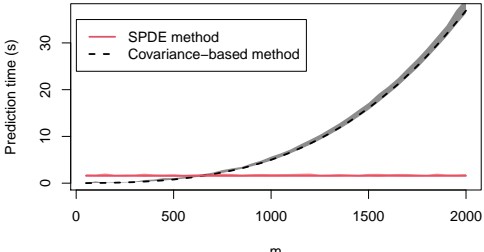

Figure 2: Left: Average RMSE with corresponding 95% pointwise confidence band for 50 reconstructions of $\mathbf{f}$ based on SPDE method with different number of basis functions $n$ (red), the corresponding RMSE for the GP model (black) and the RMSE for estimating $\mathbf{f}$ by a constant equal to the mean of the observations (blue). Right: Average computation times for the prediction of $\mathbf{f}$ as functions of the number of observations ($m$) with envelopes showing the range of times for the 50 replications.

We next fix $n = 3600$ and consider the time it takes to compute a prediction of $\mathbf{f}$ given the estimated parameters. In the right panel of Figure 2 we show this computation time as a function of the number of observations, $m$, for the baseline method and for the SPDE-based method. Also here we see that the covariance-based method is the fastest for small numbers of observations, whereas the GMRF method (that has a computational cost that scales with the number of basis functions of the SPDE approximation rather than with the number of observations) is fastest whenever $m > 600$.

## 7 Discussion

We have proposed new methods for GMRFs under linear constraints, which can greatly reduce computational costs for models with a large number of constraints. In addition, we showed how to combine these methods with the SPDE approach to allow for computationally efficient linearly constrained Gaussian process regression. Some recent papers on constrained GPs, such as [9], consider methods that are similar in spirit to those we have developed here. However, to the best of our knowledge, our methods are the first to account for sparsity, which is crucial for GMRFs.

Clearly the proposed methods will not be beneficial if the number of constraints is small. Another limitation is that the methods are only efficient if the constraints are sparse. For instance, our proposed method does not work for a sum-to-zero constraint $\sum_{i=1}^{n} X_i = 0$, which is commonly used in hierarchical models to ensure identifiability. An interesting topic for future research is to handle problems where both sparse and dense constraints are included. In that case one could combine the proposed method with a conditioning by kriging approach where the dense constraints are handled in a post-processing step as described in Section 2.

We have only considered exact methods in this work, but if one is willing to relax this requirement, an interesting alternative is the iterative Krylov subspace methods by [35]. Comparing, or combining, the proposed methods with those in [35] is thus another interesting topic for future work. A further question is whether Algorithm 2 can be improved in terms of computational cost. We have used a simple SVD implementation and for cases when the construction of the basis dominates the total cost it could worth exploring whether more modern iterative methods such as [1, 22] could be applicable. How that could be done is not clear since most such methods provide a low rank solution whereas we need the full matrix $\mathbf{V}^{\top}$. A related idea, when the exact methods presented here are not suitable, is to use subspace embeddings or sketch-based methods [19] to approximate the linear constraints. It is, however, an open question if one can get good results through such an approach, and how one could describe the accuracy of the corresponding approximated distribution.

The SPDE approach also allows for more flexible non-stationary covariance structures like the generalized Whittle–Matérn models [4]. Our proposed methods are directly applicable to these models in the Markov case (with integer smoothness), and can also be extend to the case with general smoothness by combining the constraint basis with the rational approximations in [4].

Finally, we can think of no potential negative societal impacts that this work may have, given that it is solely concerned with improving the performance of existing methods.

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
