# Supplement to *Efficient methods for Gaussian Markov random fields under sparse linear constraints*

**David Bolin**
King Abdullah University of
Science and Technology
david.bolin@kaust.edu.sa

**Jonas Wallin**
Department of Statistics,
Lund University
jonas.wallin@stat.lu.se

In this document we provide further details of the constraint basis construction as well as additional simulation results are given in Section A. We then provide the proofs of the four main theorems of the paper in Section B. Finally, Section C provides details about the conditional constrained distribution $\pi(\mathbf{x}|\mathbf{A}\mathbf{x} = \mathbf{b})$.

## A    Further details of the basis construction and simulations

In Algorithm 2 it was not explained how the reordering of the $\mathbf{A}$ matrix is done. This is illustrated in Algorithm 6 where we show how to build the sub-matrices $\{\tilde{\mathbf{A}}\}_{k=1}^{m}$.

---

**Algorithm 6** Find all non-overlapping sub-matrices

**Require:** $\mathbf{A}$ (a $k \times n$ matrix)
1: $\{\tilde{\mathbf{A}}_1, \mathbf{B}\} \leftarrow overlap(\mathbf{A})$
2: $m \leftarrow 1$
3: **while** $\mathbf{B} \neq \emptyset$ **do**
4: $\quad m \leftarrow m + 1$
5: $\quad \{\tilde{\mathbf{A}}_m, \mathbf{B}\} \leftarrow overlap(\mathbf{B})$
6: **end while**
7: Return $\{\tilde{\mathbf{A}}\}_{k=1}^{m}$

---

**Algorithm 7** $overlap(\mathbf{A})$ Find first sub-matrix

**Require:** $\mathbf{A}$ (a $k \times n$ matrix)
1: $U \leftarrow \{1\}$
2: $d \leftarrow 0$
3: $D \leftarrow id(\mathbf{A}_{U,.})$
4: **while** $1$ **do**
5: $\quad D \leftarrow id(\mathbf{A}_{U,.})$
6: $\quad U \leftarrow id\left(\left(\mathbf{A}_{.,D}\right)^{\top}\right)$
7: $\quad$ **if** $d = |U|$ **then**
8: $\quad\quad$ **break**
9: $\quad$ **end if**
10: $\quad d \leftarrow |U|$
11: **end while**
12: $\tilde{\mathbf{A}} = \mathbf{A}_{.,U}$
13: $\tilde{\mathbf{A}}^{c} = \mathbf{A}_{.,U^c}$
14: Return $\{\tilde{\mathbf{A}}, \tilde{\mathbf{A}}^{c}\}$

---

In Section 6.1 in the main article, we provided results for $\alpha = 2$ and $\alpha = 4$. In Figure 1 the corresponding results are provivded for $\alpha = 1$ and $\alpha = 3$.

## B    Proofs

In this section we prove the four main theorems of the paper.

*Proof of Theorem 1.* We first transform the density of $\mathbf{A}\mathbf{X}$ to the basis represented by $\mathbf{T}$,

$$\pi_{\mathbf{A}\mathbf{X}}(\mathbf{b}) = \pi_{\mathbf{A}\mathbf{T}^{\top}\mathbf{X}^*}(\mathbf{b}) = \pi_{\mathbf{X}_{\mathscr{C}}^*}\left(\mathbf{H}^{-1}\mathbf{b}\right)\left||\mathbf{H}|^{-1}\right| = \pi_{\mathbf{X}_{\mathscr{C}}^*}\left(\mathbf{H}^{-1}\mathbf{b}\right)|\mathbf{A}\mathbf{A}^{\top}|^{-1/2}.$$

35th Conference on Neural Information Processing Systems (NeurIPS 2021).

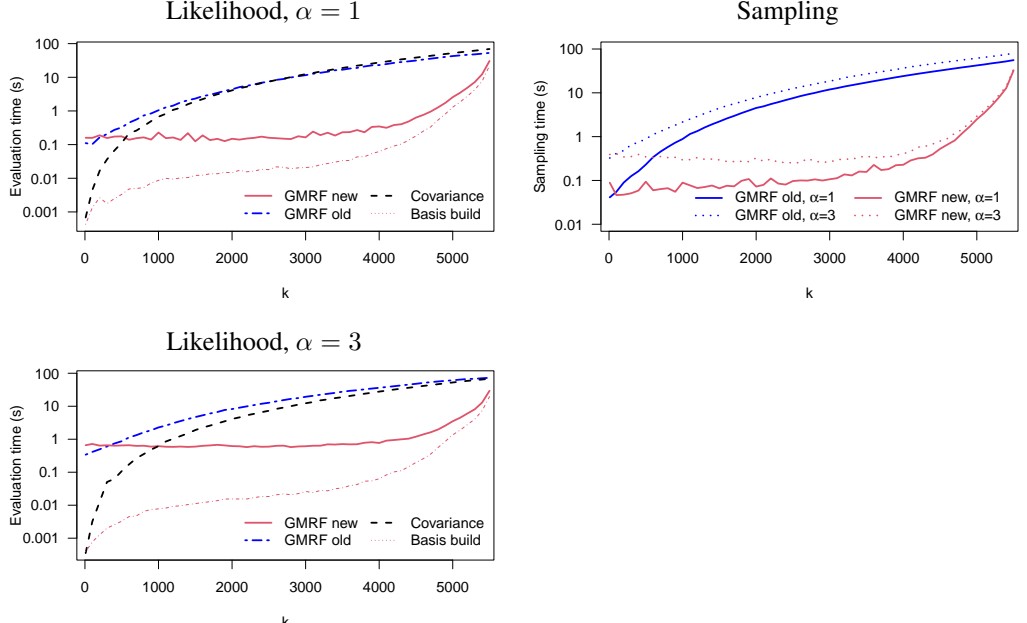

Figure 1: Average computation times (based on 10 replications) for one likelihood evaluation (left) and one sample from $\mathbf{X}|\mathbf{A}_Y\mathbf{X} = \mathbf{y}$ (top right) of the Matérn model with $\alpha = 1$ and $\alpha = 3$ as functions of the number of observations $k$. The computation times for the new GMRF method includes the time needed to construct the basis matrix $\mathbf{T}$, which also is shown separately.

In order to derive the density $\mathbf{X}^*_{\mathscr{C}}$, note that the density of $\mathbf{X}^*$ is

$$\pi_{\mathbf{X}^*_{\mathscr{C}}}(\mathbf{x}^*) = \frac{|\mathbf{Q}|^{\dagger/2}}{(2\pi)^{\frac{n-s}{2}}} \exp\left(-\frac{1}{2}Q(\mathbf{x}^*)\right),$$

where the quadratic form $Q(\mathbf{x}^*)$ is

$$
\begin{aligned}
Q(\mathbf{x}^*) &= \begin{bmatrix} \mathbf{x}^*_{\mathscr{U}} - \boldsymbol{\mu}^*_{\mathscr{U}} \\ \mathbf{x}^*_{\mathscr{C}} - \boldsymbol{\mu}^*_{\mathscr{C}} \end{bmatrix}^{\top} \begin{bmatrix} \mathbf{Q}^*_{\mathscr{U}\mathscr{U}} & \mathbf{Q}^*_{\mathscr{C}\mathscr{U}} \\ \mathbf{Q}^*_{\mathscr{C}\mathscr{U}} & \mathbf{Q}^*_{\mathscr{C}\mathscr{C}} \end{bmatrix} \begin{bmatrix} \mathbf{x}^*_{\mathscr{U}} - \boldsymbol{\mu}^*_{\mathscr{U}} \\ \mathbf{x}^*_{\mathscr{C}} - \boldsymbol{\mu}^*_{\mathscr{C}} \end{bmatrix} \\
&= (\mathbf{x}^*_{\mathscr{C}} - \boldsymbol{\mu}^*_{\mathscr{C}})^{\top} \mathbf{Q}^*_{\mathscr{C}\mathscr{C}} (\mathbf{x}^*_{\mathscr{C}} - \boldsymbol{\mu}^*_{\mathscr{C}}) + (\mathbf{x}^*_{\mathscr{C}} - \boldsymbol{\mu}^*_{\mathscr{C}})^{\top} \mathbf{Q}^*_{\mathscr{C}\mathscr{U}} (\mathbf{x}^*_{\mathscr{U}} - \boldsymbol{\mu}^*_{\mathscr{U}}) \\
&\quad + (\mathbf{x}^*_{\mathscr{U}} - \boldsymbol{\mu}^*_{\mathscr{U}})^{\top} \mathbf{Q}^*_{\mathscr{U}\mathscr{C}} (\mathbf{x}^*_{\mathscr{C}} - \boldsymbol{\mu}^*_{\mathscr{C}}) + (\mathbf{x}^*_{\mathscr{U}} - \boldsymbol{\mu}^*_{\mathscr{U}})^{\top} \mathbf{Q}^*_{\mathscr{U}\mathscr{U}} (\mathbf{x}^*_{\mathscr{U}} - \boldsymbol{\mu}^*_{\mathscr{U}}) \\
&= (\mathbf{x}^*_{\mathscr{C}} - \boldsymbol{\mu}^*_{\mathscr{C}})^{\top} \mathbf{Q}^*_{\mathscr{C}\mathscr{C}} (\mathbf{x}^*_{\mathscr{C}} - \boldsymbol{\mu}^*_{\mathscr{C}}) - (\mathbf{x}^*_{\mathscr{C}} - \boldsymbol{\mu}^*_{\mathscr{C}})^{\top} \mathbf{Q}^*_{\mathscr{C}\mathscr{U}} \mathbf{Q}^{*\dagger}_{\mathscr{U}\mathscr{U}} \mathbf{Q}^*_{\mathscr{U}\mathscr{C}} (\mathbf{x}^*_{\mathscr{C}} - \boldsymbol{\mu}^*_{\mathscr{C}}) + \\
&\quad + \left(\mathbf{x}^*_{\mathscr{U}} - \boldsymbol{\mu}^*_{\mathscr{U}} + \mathbf{Q}^{*\dagger}_{\mathscr{U}\mathscr{U}} \mathbf{Q}^*_{\mathscr{U}\mathscr{C}} (\mathbf{x}^*_{\mathscr{C}} - \boldsymbol{\mu}^*_{\mathscr{C}})\right)^{\top} \mathbf{Q}^*_{\mathscr{U}\mathscr{U}} \left(\mathbf{x}^*_{\mathscr{U}} - \boldsymbol{\mu}^*_{\mathscr{U}} + \mathbf{Q}^{*\dagger}_{\mathscr{U}\mathscr{U}} \mathbf{Q}^*_{\mathscr{U}\mathscr{C}} (\mathbf{x}^*_{\mathscr{C}} - \boldsymbol{\mu}^*_{\mathscr{C}})\right).
\end{aligned}
$$

Here we in the last step wrote the expression so that we easily can integrate out $\mathbf{X}^*_{\mathscr{U}}$ on the complement to the null space of $\mathbf{Q}^*_{\mathscr{U}\mathscr{U}}$. Doing so yields the desired result,

$$\pi(\mathbf{x}^*_{\mathscr{C}}) = \int \pi_{\mathbf{X}^*_{\mathscr{C}}}(\mathbf{x}^*) d\mathbf{x}^*_{\mathscr{U}} \propto \frac{|\mathbf{Q}|^{\dagger/2}}{|\mathbf{Q}^*_{\mathscr{U}\mathscr{U}}|^{\dagger/2}} \exp\left(-\frac{1}{2}\hat{Q}(\mathbf{x}^*_{\mathscr{C}})\right),$$

where

$$\hat{Q}(\mathbf{x}^*_{\mathscr{C}}) = (\mathbf{x}^*_{\mathscr{C}} - \boldsymbol{\mu}^*_{\mathscr{C}})^{\top} \left(\mathbf{Q}^*_{\mathscr{C}\mathscr{C}} - \mathbf{Q}^*_{\mathscr{C}\mathscr{U}} \mathbf{Q}^{*\dagger}_{\mathscr{U}\mathscr{U}} \mathbf{Q}^*_{\mathscr{U}\mathscr{C}}\right) (\mathbf{x}^*_{\mathscr{C}} - \boldsymbol{\mu}^*_{\mathscr{C}}).$$

$\square$

To prove Theorem 2 we need the following lemma.

**Lemma 1.** *Under Assumption 1 one has* $rank\left(\mathbf{Q}^*_{\mathscr{C}\mathscr{C}}\right) = k - k_0$ *and* $rank\left(\mathbf{Q}^*_{\mathscr{U}\mathscr{U}}\right) = n - s - (k - k_0)$.

*Proof.* We have $rank(\mathbf{Q}^*) = rank(\mathbf{Q}) = n - s$ since $\mathbf{T}$ is orthonormal matrix. Further, using the eigen-decomposition of $\mathbf{Q}$ we can express $\mathbf{Q}^*$ as

$$\mathbf{Q}^* = \begin{bmatrix} \mathbf{T}_{\mathscr{C}} \\ \mathbf{T}_{\mathscr{U}} \end{bmatrix} \begin{bmatrix} \mathbf{E}_{0^c} \\ \mathbf{E}_0 \end{bmatrix} \begin{bmatrix} \boldsymbol{\Lambda} & \mathbf{0} \\ \mathbf{0} & \mathbf{0} \end{bmatrix} \begin{bmatrix} \mathbf{E}_{0^c} \\ \mathbf{E}_0 \end{bmatrix}^{\top} \begin{bmatrix} \mathbf{T}_{\mathscr{C}} \\ \mathbf{T}_{\mathscr{U}} \end{bmatrix}^{\top},$$

where $\mathbf{\Lambda}$ is a diagonal matrix with the non-zero eigenvalues of $\mathbf{Q}$. Since $rank(\mathbf{AE}_0) = k_0$ it follows that also $rank(\mathbf{T}_{\mathscr{C}}\mathbf{E}_0) = k_0$ and $rank(\mathbf{T}_{\mathscr{U}}\mathbf{E}_0) = s - k_0$. By Theorem 4.3.28 of [HJ13] there exists an eigenvector, $\mathbf{e}$, of $\mathbf{Q}^*_{\mathscr{U}\mathscr{U}}$ that has a corresponding eigenvalue 0 if and only if $\mathbf{Q}^*_{\mathscr{U}\mathscr{U}}\mathbf{e} = \mathbf{0}$ and $\mathbf{Q}^*_{\mathscr{C}\mathscr{U}}\mathbf{e} = \mathbf{0}$. By construction, any vector constructed by the linear span of $\mathbf{T}_{\mathscr{U}}\mathbf{E}_0$ satisfies this requirement, and no other vector does. Hence, the rank of $\mathbf{Q}^*_{\mathscr{U}\mathscr{U}}$ is $n - k - (s - k_0)$ and the rank of $\mathbf{Q}^*_{\mathscr{C}\mathscr{C}}$ is $k - k_0$. $\qquad\square$

*Proof of Theorem 2.* To derive the distribution we note that the conditional distribution of $\mathbf{X}^*_{\mathscr{U}}|\mathbf{X}^*_{\mathscr{C}}$ is proportional to $\exp(-\frac{1}{2}Q(\mathbf{x}^*))$, where

$$
\begin{aligned}
Q(\mathbf{x}^*) &= \begin{bmatrix} \mathbf{x}^*_{\mathscr{U}} - \boldsymbol{\mu}^*_{\mathscr{U}} \\ \mathbf{x}^*_{\mathscr{C}} - \boldsymbol{\mu}^*_{\mathscr{C}} \end{bmatrix}^{\top} \begin{bmatrix} \mathbf{Q}^*_{\mathscr{U}\mathscr{U}} & \mathbf{Q}^*_{\mathscr{C}\mathscr{U}} \\ \mathbf{Q}^*_{\mathscr{C}\mathscr{U}} & \mathbf{Q}^*_{\mathscr{C}\mathscr{C}} \end{bmatrix} \begin{bmatrix} \mathbf{x}^*_{\mathscr{U}} - \boldsymbol{\mu}^*_{\mathscr{U}} \\ \mathbf{x}^*_{\mathscr{C}} - \boldsymbol{\mu}^*_{\mathscr{C}} \end{bmatrix} \\
&= (\mathbf{x}^*_{\mathscr{U}} - \boldsymbol{\mu}^*_{\mathscr{U}})^{\top}\,\mathbf{Q}^*_{\mathscr{U}\mathscr{U}}\,(\mathbf{x}^*_{\mathscr{U}} - \boldsymbol{\mu}^*_{\mathscr{U}}) + 2\,(\mathbf{x}^*_{\mathscr{U}} - \boldsymbol{\mu}^*_{\mathscr{U}})^{\top}\,\mathbf{Q}^*_{\mathscr{U}\mathscr{C}}\,(\mathbf{x}^*_{\mathscr{C}} - \boldsymbol{\mu}^*_{\mathscr{C}}) + C,
\end{aligned}
$$

where $C$ is a constant independent of $\mathbf{x}^*_{\mathscr{U}}$. Now, since $\mathbf{Q}^*_{\mathscr{U}\mathscr{U}}\,(\mathbf{Q}^*_{\mathscr{U}\mathscr{U}})^{\dagger}\,\mathbf{Q}^*_{\mathscr{U}\mathscr{C}} = \mathbf{Q}^*_{\mathscr{U}\mathscr{C}}$, the quadratic form $Q(\mathbf{x}^*)$ can be written as a constant plus $\mathbf{v}^{\top}\mathbf{Q}^*_{\mathscr{U}\mathscr{C}}\mathbf{v}$, where

$$
\mathbf{v} = \mathbf{x}^*_{\mathscr{U}} - \boldsymbol{\mu}^*_{\mathscr{U}} + \mathbf{Q}^{*\dagger}_{\mathscr{U}\mathscr{U}}\mathbf{Q}^*_{\mathscr{U}\mathscr{C}}(\mathbf{x}^*_{\mathscr{C}} - \boldsymbol{\mu}^*_{\mathscr{C}}).
$$

Hence

$$
\mathbf{X}^*_{\mathscr{U}}|\mathbf{X}^*_{\mathscr{C}} = \mathbf{b}^* \sim \mathcal{N}_C\left(\mathbf{Q}^*_{\mathscr{U}\mathscr{U}}\left(\boldsymbol{\mu}^*_{\mathscr{U}} - \mathbf{Q}^{*\dagger}_{\mathscr{U}\mathscr{U}}\mathbf{Q}^*_{\mathscr{U}\mathscr{C}}(\mathbf{b}^* - \boldsymbol{\mu}^*_{\mathscr{C}})\right), \mathbf{Q}^*_{\mathscr{U}\mathscr{U}}\right). \tag{10}
$$

Combining this with the fact that $\mathbf{X} = \mathbf{T}^{\top}\mathbf{X}^*$ gives the desired expression for the distribution of $\mathbf{X}|\mathbf{AX} = \mathbf{b}$. Finally, since $\mathbf{T}$ is orthonormal, the rank of $\mathbf{Q}_{X|b}$ is the same as the rank of $\mathbf{Q}_{\mathscr{U}\mathscr{U}}$, and the result follows from Lemma 1. $\qquad\square$

We start by proving Theorem 4 as we will use result from that proof in the proof of Theorem 3.

*Proof of Theorem 4.* First, note that the density

$$
\pi_{\mathbf{Y}|\mathbf{X}^*_{\mathscr{U}},\mathbf{X}^*_{\mathscr{C}}}(\mathbf{y}|\mathbf{x}^*_{\mathscr{U}},\mathbf{b}^*) = \frac{1}{(2\pi)^{\frac{m}{2}}\sigma^m_Y}\exp\left(-\frac{1}{2\sigma^2_Y}\left(\mathbf{y} - \mathbf{B}^*\begin{bmatrix}\mathbf{b}^* \\ \mathbf{x}^*_{\mathscr{U}}\end{bmatrix}\right)\left(\mathbf{y} - \mathbf{B}^*\begin{bmatrix}\mathbf{b}^* \\ \mathbf{x}^*_{\mathscr{U}}\end{bmatrix}\right)\right), \tag{11}
$$

can, as a function of $\mathbf{x}^*_{\mathscr{U}}$, be written as

$$
\pi_{\mathbf{Y}|\mathbf{X}^*_{\mathscr{C}},\mathbf{X}^*_{\mathscr{U}}}(\mathbf{y}|\mathbf{b}^*,\mathbf{x}^*_{\mathscr{U}}) \propto \exp\left(-\frac{\mathbf{x}^{*\top}_{\mathscr{U}}\mathbf{B}^{*\top}_{\mathscr{U}}\mathbf{B}^*_{\mathscr{U}}\mathbf{x}^*_{\mathscr{U}}}{2\sigma^2_Y} + \frac{\mathbf{y}^{*\top}\mathbf{B}^*_{\mathscr{U}}\mathbf{x}^*_{\mathscr{U}}}{\sigma^2_Y}\right).
$$

Further, from (10), we have that, as a function of $\mathbf{x}^*_{\mathscr{U}}$,

$$
\pi_{\mathbf{X}^*_{\mathscr{U}}|\mathbf{X}^*_{\mathscr{C}}}(\mathbf{x}^*_{\mathscr{U}}|\mathbf{b}^*) \propto \exp\left(-\frac{1}{2}\left(\mathbf{x}^*_{\mathscr{U}} - \widetilde{\boldsymbol{\mu}}^*_{\mathscr{U}}\right)^{\top}\mathbf{Q}^*_{\mathscr{U}\mathscr{U}}\left(\mathbf{x}^*_{\mathscr{U}} - \widetilde{\boldsymbol{\mu}}^*_{\mathscr{U}}\right)\right),
$$

where $\widetilde{\boldsymbol{\mu}}^*_{\mathscr{U}} = \boldsymbol{\mu}^*_{\mathscr{U}} - \mathbf{Q}^{*\dagger}_{\mathscr{U}\mathscr{U}}\mathbf{Q}^*_{\mathscr{U}\mathscr{C}}(\mathbf{b}^* - \boldsymbol{\mu}^*_{\mathscr{C}})$. Since $\pi_{\mathbf{X}^*_{\mathscr{U}}|\mathbf{Y},\mathbf{X}^*_{\mathscr{C}}}(\mathbf{x}^*_{\mathscr{U}}|\mathbf{y},\mathbf{b}^*)$ is proportional to $\pi_{\mathbf{Y}|\mathbf{X}^*_{\mathscr{C}},\mathbf{X}^*_{\mathscr{U}}}(\mathbf{y}|\mathbf{b}^*,\mathbf{x}^*_{\mathscr{U}})\pi(\mathbf{x}^*_{\mathscr{U}}|\mathbf{b}^*)$, it follows that

$$
\begin{aligned}
\pi_{\mathbf{X}^*_{\mathscr{U}}|\mathbf{Y},\mathbf{X}^*_{\mathscr{C}}}(\mathbf{x}^*_{\mathscr{U}}|\mathbf{y},\mathbf{b}^*) &\propto \exp\left(-\frac{1}{2}\mathbf{x}^{*\top}_{\mathscr{U}}\frac{\mathbf{B}^{*\top}_{\mathscr{U}}\mathbf{B}^*_{\mathscr{U}}}{\sigma^2_Y}\mathbf{x}^*_{\mathscr{U}} + \left(\frac{\mathbf{B}^{*\top}_{\mathscr{U}}\mathbf{y}^*}{\sigma^2_Y}\right)^{\top}\mathbf{x}^*_{\mathscr{U}}\right). \\
&\qquad \exp\left(-\frac{1}{2}\mathbf{x}^{*\top}_{\mathscr{U}}\mathbf{Q}^*_{\mathscr{U}\mathscr{U}}\mathbf{x}^*_{\mathscr{U}} + \left(\mathbf{Q}^*_{\mathscr{U}\mathscr{U}}\widetilde{\boldsymbol{\mu}}^*_{\mathscr{U}}\right)^{\top}\mathbf{x}^*_{\mathscr{U}}\right) \\
&\qquad\qquad \propto \exp\left(-\frac{1}{2}\left(\mathbf{x}^*_{\mathscr{U}} - \widehat{\boldsymbol{\mu}}^*_{\mathscr{U}}\right)^{\top}\widehat{\mathbf{Q}}^*_{\mathscr{U}\mathscr{U}}\left(\mathbf{x}^*_{\mathscr{U}} - \widehat{\boldsymbol{\mu}}^*_{\mathscr{U}}\right)\right).
\end{aligned}
$$

Finally, using the relation $\mathbf{X} = \mathbf{T}^{\top}\mathbf{X}^*$ completes the proof. $\qquad\square$

*Proof of Theorem 3.* First note that $\pi_{\mathbf{Y}|\mathbf{AX}}(\mathbf{y}|\mathbf{b}) = \pi_{\mathbf{Y}|\mathbf{X}^*_{\mathscr{C}}}(\mathbf{y}|\mathbf{b}^*)$ and

$$
\begin{aligned}
\pi_{\mathbf{Y}|\mathbf{X}^*_{\mathscr{C}}}(\mathbf{y}|\mathbf{b}^*) &= \int \pi_{\mathbf{X}^*_{\mathscr{U}},\mathbf{Y}|\mathbf{X}^*_{\mathscr{C}}}(\mathbf{x}^*_{\mathscr{U}},\mathbf{y}|\mathbf{b}^*)\,d\mathbf{x}^*_{\mathscr{U}} \\
&= \int \pi_{\mathbf{Y}|\mathbf{X}^*_{\mathscr{U}},\mathbf{X}^*_{\mathscr{C}}}(\mathbf{y}|\mathbf{x}^*_{\mathscr{U}},\mathbf{b}^*)\pi_{\mathbf{X}^*_{\mathscr{U}}|\mathbf{X}^*_{\mathscr{C}}}(\mathbf{x}^*_{\mathscr{U}}|\mathbf{b}^*)\,d\mathbf{x}^*_{\mathscr{U}}. \quad (12)
\end{aligned}
$$

The goal is now to derive an explicit form of the density by evaluating the integral in (12). By the expressions in the proof of Theorem 4 we have

$$
\begin{aligned}
\pi_{\mathbf{Y}|\mathbf{X}^*_{\mathscr{U}},\mathbf{X}^*_{\mathscr{C}}}(\mathbf{y}|\mathbf{x}^*_{\mathscr{U}},\mathbf{b}^*)\pi_{\mathbf{X}^*_{\mathscr{U}}|\mathbf{X}^*_{\mathscr{C}}}(\mathbf{x}^*_{\mathscr{U}}|\mathbf{b}^*) &= \exp\left(-\frac{1}{2}\mathbf{x}^{*\top}_{\mathscr{U}}\frac{\mathbf{B}^{*\top}_{\mathscr{U}}\mathbf{B}^*_{\mathscr{U}}}{\sigma_Y^2}\mathbf{x}^*_{\mathscr{U}} + \left(\frac{\mathbf{B}^{*\top}_{\mathscr{U}}\mathbf{y}^*}{\sigma_Y^2}\right)^\top \mathbf{x}^*_{\mathscr{U}}\right)\cdot \\
&\quad \exp\left(-\frac{1}{2}\mathbf{x}^{*\top}_{\mathscr{U}}\mathbf{Q}^*_{\mathscr{U}\mathscr{U}}\mathbf{x}^*_{\mathscr{U}} + \left(\mathbf{Q}^*_{\mathscr{U}\mathscr{U}}\widetilde{\boldsymbol{\mu}}^*_{\mathscr{U}}\right)^\top \mathbf{x}^*_{\mathscr{U}}\right)\cdot \\
&\quad \frac{|\mathbf{Q}^*_{\mathscr{U}\mathscr{U}}|^{\dagger/2}}{(2\pi)^{c_0}\sigma_Y^m}\exp\left(-\frac{1}{2}\left[\frac{\mathbf{y}^{*\top}\mathbf{y}^*}{\sigma_Y^2} + \widetilde{\boldsymbol{\mu}}^{*\top}_{\mathscr{U}}\mathbf{Q}^*_{\mathscr{U}\mathscr{U}}\widetilde{\boldsymbol{\mu}}^*_{\mathscr{U}}\right]\right) \\
&= \pi_{\mathbf{X}^*_{\mathscr{U}}|\mathbf{Y},\mathbf{X}^*_{\mathscr{C}}}(\mathbf{x}^*_{\mathscr{U}}|\mathbf{y},\mathbf{b}^*)\frac{\exp\left(\frac{1}{2}\widehat{\boldsymbol{\mu}}^{*\top}_{\mathscr{U}}\widehat{\mathbf{Q}}^*_{\mathscr{U}\mathscr{U}}\widehat{\boldsymbol{\mu}}^*_{\mathscr{U}}\right)}{|\widehat{\mathbf{Q}}^*_{\mathscr{U}\mathscr{U}}|^{\dagger/2}}\cdot \\
&\quad \frac{|\mathbf{Q}^*_{\mathscr{U}\mathscr{U}}|^{\dagger/2}}{(2\pi)^{c_1}\sigma_Y^m}\exp\left(-\frac{1}{2}\left[\frac{\mathbf{y}^{*\top}\mathbf{y}^*}{\sigma_Y^2} + \boldsymbol{\mu}^{*\top}_{\mathscr{U}}\mathbf{Q}^*_{\mathscr{U}\mathscr{U}}\boldsymbol{\mu}^*_{\mathscr{U}}\right]\right),
\end{aligned}
$$

where $c_0$ and $c_1$ are positive constants. Inserting this expression in (12) and evaluating the integral, where one notes that $\pi_{\mathbf{X}^*_{\mathscr{U}}|\mathbf{Y},\mathbf{X}^*_{\mathscr{C}}}(\mathbf{x}^*_{\mathscr{U}}|\mathbf{y},\mathbf{b}^*)$ integrates to one, gives the desired result. $\qquad\square$

## C   Conditional constrained distribution

In order to derive the conditional density $\pi(\mathbf{x}|\mathbf{Ax}=\mathbf{b})$ in (3) we will use what is known as the disintegration technique. The proof is built on the results in [CP97], which has the following definition.

**Definition 1.** *Let $(\mathcal{X},\mathcal{A},\lambda)$ and $(\mathcal{T},\mathcal{B},\mu)$ be two measure spaces with $\sigma$-finite measures $\lambda$ and $\mu$. The measure $\lambda$ has a disintegration $\{\lambda_b\}$ with respect to the measurable map $A:(\mathcal{X},\mathcal{A})\to(\mathcal{T},\mathcal{B})$ and the measure $\mu$, or a $(A(x),\mu)-$disintegration if:*

*(i) $\lambda_b$ is a $\sigma$-finite measure on $\mathcal{A}$ such that $\lambda_b(A(x)\neq b)=0,$ for $\mu-$almost all $b$,*

*and, for each non-negative measurable function $f$ on $\mathcal{X}$:*

*(ii) $b\to\int f d\lambda_b$ is measurable.*

*(iii) $\int f d\lambda = \int\int f d\lambda_b d\mu$.*

In the following theorem, we use the notation from Appendix B and let $\lambda_n$ denote the Lebesgue measure on $\mathbb{R}^n$. Further, we define $\lambda_{\mathscr{U}}$ as the image measure of the projection onto the image of $\mathbf{A}$ (which is not $\sigma$-finite), and $\lambda_{\mathscr{C}}$ as the image measure of the projection onto the null-space of $\mathbf{A}$.

**Theorem 6.** *Let $\mathbf{X}$ be a multivariate random variable with distribution $\mathbb{P}$ on $(\mathbb{R}^n,\mathcal{B}(\mathbb{R}^n))$, where $\mathbb{P}$ has density $\pi(\mathbf{x})$ with respect to $\lambda_n$. Then the random variable $\mathbf{X}|\mathbf{AX}=\mathbf{b}$ has density*

$$
\pi(\mathbf{x}|\mathbf{Ax}=\mathbf{b}) = \frac{\mathbb{I}(\mathbf{Ax}=\mathbf{b})|\mathbf{AA}^\top|^{-1/2}\pi(\mathbf{x})}{\pi_{\mathbf{AX}}(\mathbf{b})},
$$

*with respect to the measure $\mathcal{L}_{\mathbf{b}}(\cdot) = \lambda_{\mathscr{U}}(\cdot\cap\{x:\mathbf{Ax}=\mathbf{b}\})$ on $(\mathbb{R}^n,\mathcal{B}(\mathbb{R}^n))$.*

The proof is based on the following lemma.

**Lemma 2.** *The measure $\mathcal{L}_{\mathbf{b}}(\cdot)$ is the $(\mathbf{A},\mathcal{L}^k)$-disintegration of the Lebesgue measure $\lambda_n$.*

*Proof.* Thus we need to show that (i), (ii), and (iii) of Definition 1 holds. Clearly, (i) follows immediate from $\cdot \cap \{x : \mathbf{A}\mathbf{x} = \mathbf{b}\}$. To show (ii), note that

$$\int f d\mathcal{L}_{\mathbf{b}} = \int f\left(\mathbf{T}^{\top}\mathbf{x}^*\right) \mathbb{I}_{\mathbf{A}\mathbf{T}^{\top}\mathbf{x}^*=\mathbf{b}} \left(d\mathbf{x}^*\right) d\mathbf{x}^*_{\mathcal{U}}$$

$$= ||\mathbf{H}|| \int_{\{\mathbf{x}^*:\mathbf{x}^*_{\mathcal{C}}=\mathbf{H}^{-1}\mathbf{b}\}} f^*(\mathbf{x}^*_{\mathcal{C}}, \mathbf{x}^*_{\mathcal{U}}) d\mathbf{x}^*_{\mathcal{U}} = ||\mathbf{H}|| \int f^*(\mathbf{H}^{-1}\mathbf{b}, \mathbf{x}^*_{\mathcal{U}}) d\mathbf{x}^*_{\mathcal{U}},$$

where $\mathbf{H} = \left(\mathbf{A}\mathbf{T}^{\top}\right)_{\mathcal{C}\mathcal{C}}$ as defined in Section 3.2, $||\mathbf{H}||$ denotes the absolute value of the determinant of $\mathbf{H}$, and $f^*(\mathbf{x}^*) = f(\mathbf{T}^{\top}\mathbf{x})$. Since $f$ is a measurable function it follows by Tonelli Theorem [Pol02] that above partial integral is measurable. Finally, to show (iii), we continue from the equation above and get

$$\iint f d\mathcal{L}_{\mathbf{b}} d\mathbf{b} = \int ||\mathbf{H}|| \int f^*(\mathbf{H}^{-1}\mathbf{b}, \mathbf{x}^*_{\mathcal{U}}) d\mathbf{x}^*_{\mathcal{U}} d\mathbf{b}$$

$$= ||\mathbf{H}|| \, |\mathbf{H}|^{-1}| \iint f^*(\mathbf{b}^*, \mathbf{x}^*_{\mathcal{U}}) d\mathbf{x}^*_{\mathcal{U}} d\mathbf{b}^* = \int f d\lambda_n.$$

$\square$

*Proof of Theorem 6.* By Lemma 2 above and Theorem 3 (v) in [CP97] it follows that the random variable has density

$$\pi(\mathbf{x}|\mathbf{A}\mathbf{X} = \mathbf{b}) = \frac{\pi(\mathbf{x})}{\mathcal{L}_{\mathbf{b}}\pi(\mathbf{x})} = \frac{\pi(\mathbf{x})}{\pi_{\mathbf{A}\mathbf{X}}(\mathbf{b}) \, ||\mathbf{H}||} = \frac{|\mathbf{A}\mathbf{A}^{\top}|^{-1/2}\pi(\mathbf{x})}{\pi_{\mathbf{A}\mathbf{X}}(\mathbf{b})},$$

a.e. with respect to $\mathcal{L}_{\mathbf{b}}$. Finally, it holds that $\pi(\mathbf{x}|\mathbf{A}\mathbf{X} = \mathbf{b}) = \mathbb{I}(\mathbf{A}\mathbf{x} = \mathbf{b})\,\pi(\mathbf{x}|\mathbf{A}\mathbf{X} = \mathbf{b})$ a.e. since $\mathcal{L}_{\mathbf{b}}(\cdot) = \lambda_{\mathcal{U}}(\cdot \cap \{\mathbf{x} : \mathbf{A}\mathbf{x} = \mathbf{b}\})$. $\square$