# OpenReview forum: "Efficient methods for Gaussian Markov random fields under sparse linear constraints"
_NeurIPS.cc/2021/Conference — NeurIPS 2021 Poster_

### Official Review · Reviewer_rVma · 2021-07-14

**Rating:** 7
**Confidence:** 3

**Summary:**

This paper introduces efficient methods for GP inference under the Markov assumption when conditioning on sparse linear combinations. The approach relies on astute changes of basis. It assumes that the matrix A representing the linear mapping involved in the conditioning can be represented in block diagonal form, enabling numerical savings in the resulting SVD. Decomposing the finite dimensional space of interest appropriately also enables enjoying elegant and practical results, as illustrated by their application to GP/GMRF modelling under hard and soft (sparse) constraints. A noticeable fact about the presented approach ist hat it also applies to intrinsic GMRFs.

**Limitations And Societal Impact:**

Ok

**Main Review:**

This is a neat contribution touching upon computational spatial statistics and Gaussian Process modelling. The main approach is quite simple yet potentially very useful in GP applications, as illustrated at the end of the paper. The paper is written in a quite formal way, which I personally apreciate, but could make the reading not so easy to part of ist potential readership. To take a simple example, while it is of interest to tackle the intrinsic case it might be valuable to motivate more in which circumstances it could be of practical use. Also, there is room for better motivating the relevance of the sparse constraint assumption: when it practice is this assumption met / reasonable to assume vs when is it not; also in examples in could be worth stressing more to what extent the constraints are sparse and how the matrix A could actually be refactored in simpler form. What if the constraints are not exactly sparse (but block diagonal up to very small terms outside of blocks)?

A few additional comments along the pages follow:
•	One speaks of fields but from Equation (1) one works with a random vector (n coordinates).
•	I am not fond oft he hat notation as it suggests estimation. Better use tilda for instance?
•	In what sense is Q^-1 unbounded? You mean Q^-1 has 0 eigenvalues, right?
•	What is k0 in Assumption 1? Is it related to something else or named on the spot?
•	Should curly C and U be vectors rather than sets as the order matters?
•	Unless the number of authors is very large I would avoid „et al.“ in the references

Post-discussion: thank you to the authors for their responses. I maintain my score.

**Time Spent Reviewing:**

2.5

---

> ### Author Response · Authors · 2021-08-10
> **Response to Reviewer rVma**
>
> Thank you for your comments, we are glad that you like the manuscript. Please find our answers to your comments below.
>
> >To take a simple example, while it is of interest to tackle the intrinsic case it might be valuable to motivate more in which circumstances it could be of practical use. Also, there is room for better motivating the relevance of the sparse constraint assumption: when it practice is this assumption met / reasonable to assume vs when is it not; also in examples in could be worth stressing more to what extent the constraints are sparse and how the matrix A could actually be refactored in simpler form. What if the constraints are not exactly sparse (but block diagonal up to very small terms outside of blocks)?
>
> We will add this to the introduction. Intrinsic fields are often used as priors in Bayesian models, where one for example may want to have a prior that is invariant to the addition to a constant, so that no prior assumption is made on the expected value of the field. We have some references to applications of such models in the introduction, but we will add some more to further illustrate the importance.
>
> Regarding the sparse constraints, they often occur when the constraints in some sense are local (i.e., only affecting a small neighbourhood). This is, for example, the case for constraints on derivatives, constraints on sums of nearby observations, or constraints on sums of  independent Gaussian random fields. However, the local observations must not be too dense. For instance, if one applies the true gradient operator as presented in [1] this a local operation but as the "observations" are completely dense this would result in a non-sparse method. We overcome this by performing the approximation of not taking the gradient at every node location but instead taking only every third row and column to apply the constraint. This approximation converges to the true operator as one makes the grid finer. That is thus an example of where we can make a sparse approximation to be able to use the method in more general circumstances.
>
> [1] Carl Jidling et al. “Linearly constrained Gaussian processes”. In: Advances in Neural Information Processing Systems 30. Ed. by I. Guyon et al. Curran Associates, Inc., 2017, pp. 1215–1224
>
> >One speaks of fields but from Equation (1) one works with a random vector (n coordinates)
>
> Here $n$ is the number of observations, the term Random field is just a random variable over an arbitrary domain (although typically $\mathbb{R}^d$). For instance if the domain is $\mathbb{R}^d$ we would have $X_1=X(s_1),X_2=X(s_2),\ldots, X_n = X(s_n)$ where $s_i  \in \mathbb{R}^d$, and $X$ is a Gaussian random field. Alternatively, for GMRFs the domain is typically discrete, so the random field is, as you say, just a multivariate normal random vector.
>
> >I am not fond oft he hat notation as it suggests estimation. Better use tilda for instance?
>
> We acknowledge that the notation is somewhat heavy and would be happy to find a better one. We already used tilde to denote covariances and means conditioning on $AX=b$, and hat for conditioning on $BX=y$. However, we will think further about if we can find better notations and we will also add a small table explaning all notation.
>
> > In what sense is $Q^{-1}$ unbounded? You mean $Q^{-1}$ has 0 eigenvalues, right?
>
> Yes, $Q^{-1}$ is unbounded when $Q$ has at least one zero eigenvalue. This is the case for intrinsic models like random walks that satisfy $Q1=0$. $Q^{-1}$ has a zero eigenvalues as soon as we add a hard constraint. In this article we handle both problems.
>
> >What is k0 in Assumption 1?  Is it related to something else or named on the spot?
>
> It is just the notation we use for that matrix rank, which specifically is the "amount" of the null-space that we have information about through the observations. So it is not related to something else. We will clarify this.
>
> > Should curly C and U be vectors rather than sets as the order matters?
>
> Yes, that is probably clearer. We will clarify that, for example, $v_{\mathcal{C}}$ means the first $k$ elements of the vector $v$ in the standard order.
>
> >Unless the number of authors is very large I would avoid „et al.“ in the references
>
> We agree, however, this seems to be the way the NeurIPS bib style works. We have not added any "et al" manually.

---

### Official Review · Reviewer_VGtS · 2021-07-16

**Rating:** 6
**Confidence:** 4

**Summary:**

This paper formulates a class of computationally efficient methods for Gaussian Markov Random field (GMRF) models with linear constraints. The main idea behind the methods relies on a transformation of basis such that the constraints are easier to enforce upon transformation. The methods outperform existing alternative formulations of GMRFs under scenarios involving large numbers of sparse constraints. Finally, a GMRF for linearly constrained Gaussian process regression is derived by utilizing the proposed methods in conjunction with the stochastic partial differential equation (SPDE) approach.


**Limitations And Societal Impact:**

The authors claim that the rows of A can be split into m sub-matrices (blocks) such that each pair of blocks have no common non-zero columns; and then the SVD sub-matrices can be computed separately. I am wondering if the authors have investigated other approaches for the same purpose (e.g., block decomposition methods such as the block SVD power method)? If so, I would encourage the authors to briefly discuss this in their response.

The specifics of the covariance-based method are not explained nor is it disclosed which particular algorithm was used. Moreover, no comparison to sparse inverse covariance estimation methods has been performed. In that regard, the proposed GMRF methods are compared solely to other exact methods. In some cases, an approximation might exhibit sufficient performance, while being more efficient.

A more thorough experimental evaluation is needed. The main limitations in this regard are summarized as follows:
There is an analysis on how the number of observations or basis functions affect the RMSE and the sampling/prediction time. However, there is no analysis on how any of the proposed GMRF variants performs under different levels of sparsity. Has such an experiment been conducted? If so, I would encourage the authors to elaborate on their observations.

Too many assumptions are made in the synthetic data generation process. I am wondering if the experiments with GMRF with hard constraints have been repeated for other values of $k^2$, $\phi$, $\alpha$  (other than the [0.5, 1, 2] setting)? Moreover, in the experiments with the constrained Gaussian processes, $\alpha$ and $\sigma_2$ are simply fixed to $0.01$ and $10^{-4}$, respectively, with no explanation.

No experiments on real-world regression datasets have been conducted, particularly in the  case of the GMRF variant for Gaussian process regression.


**Main Review:**

Originality: The efficient GMRF methods introduced in this work perform transformation of basis that separates constrained and unconstrained subspaces using SVD, thereby bypassing sparsity challenges and allowing for effective enforcement of the constraints in the transformed space. Nevertheless, the idea of precision matrix decomposition through SVD, or many other decomposition methods for that matter has been extensively studied. In that sense, this work appears to be a combination of well-known practices with sparse gaussian random field models. In other words, the main originality lies in using matrix decomposition in a linearly constrained GMRF setting, however, the idea of precision matrix decomposition in general is well-studied. Also, related studies involving efficient GMRFs based on SVD/eigenvalue/Cholesky decomposition do not seem to be given credit.

Quality: The proposed GMRF methods are (1) introduced primarily for GMRFs with sparse hard (linear) constraints, (2) extended to cases of both hard and soft constraints, and (3) derived and adapted for constrained Gaussian processes. Each variant is technically sound and well supported by theoretical claims involving reasonable assumptions on the positive semi-definiteness of the precision matrix Q (typically assumed in similar settings) and likelihood derivations. Also, using the change of basis, alternative formulations of the distributions of AX and X|AX = b are derived, suitable for efficient sampling and likelihood calculation.

Clarity: The paper is generally well written and organized. The notation is clear, not difficult to follow, and consistent throughout the paper.

Significance: The problem considered in this paper is of considerable importance as efficient sampling and inference with GMRFs as well as constrained Gaussian processes find application in a broad range of areas. On the other hand, I believe that the narrow (i.e., quite specific) settings of the experimental design (particularly in the data generation process) limit the contribution’s significance.

UPDATE AFTER THE REBUTTAL: In their response, the authors have addressed the majority of the comments that I have raised. The authors have also conducted the additional assessment of the GMRF variants under different levels of sparsity that I suggested, summarized the results in the response and will reflect them in the appendix of the paper. Certain points around (1) the contribution's significance and (2) the relation between the SVD-based constraint basis construction and other decomposition methods remain unresolved; however, the authors claim that further clarifications on these points will be included in the revised version of the paper. Therefore, I decided to increase my initially assigned score from 5 to 6 (Marginally above the acceptance threshold).

**Time Spent Reviewing:**

10

---

> ### Author Response · Authors · 2021-08-10
> **Response to Reviewer VGtS**
>
> Thank you for your comments and for taking the time to go through the paper so carefully, please find our responses to the comments below.
>
> >Nevertheless, the idea of precision matrix decomposition through SVD, or many other decomposition methods for that matter has been extensively studied. In that sense, this work appears to be a combination of well-known practices with sparse gaussian random field models. In other words, the main originality lies in using matrix decomposition in a linearly constrained GMRF setting, however, the idea of precision matrix decomposition in general is well-studied. Also, related studies involving efficient GMRFs based on SVD/eigenvalue/Cholesky decomposition do not seem to be given credit.
>
> Yes, precision matrix decomposition is a well-studied area; however, we have not been able to find any methods using SVD/eigenvalue/Cholesky decomposition for sparse precision matrices under hard constraints. But in any case, we will add more general references to such methods for non-constrained models in the introduction.
>
> >Significance: The problem considered in this paper is of considerable importance as efficient sampling and inference with GMRFs as well as constrained Gaussian processes find application in a broad range of areas. On the other hand, I believe that the narrow (i.e., quite specific) settings of the experimental design (particularly in the data generation process) limit the contribution’s significance.
>
> The method is as general as possible for GMRFs under sparse constraints, so we assume that you are referring to that the examples are quite specific? We choose the first example as it is a clear example of how the method works. We choose the second example since there is currently a large interest in developing Gaussian processes (GP) under constraints where the constraints are chosen so that the GP satisfies some physical equation (there is a large number of references to such papers in the introduction). To our knowledge we are the first that have been able to use these types of models together with sparse precision matrices. Thus the applications are, as far as we know, novel since there have been no methods to deal with them before. Since GMRFs are so popular models, we believe that the introduction of our methods have the potential to make them popular also for constrained GP models. We will more clearly state this.
>
>
> >The authors claim that the rows of A can be split into m sub-matrices (blocks) such that each pair of blocks have no common non-zero columns; and then the SVD sub-matrices can be computed separately. I am wondering if the authors have investigated other approaches for the same purpose (e.g., block decomposition methods such as the block SVD power method)? If so, I would encourage the authors to briefly discuss this in their response.
>
> We should acknowledge that we are not experts on matrix decomposition methods, but rather use them for statistical applications. We have searched for various SVD methods and other basis building operations to try to find something that does what we need, but have not been able find any suitable algorithms. To our understanding the block power methods is used to compute the $k-$largest singular value basis? We have not been able to find a method that computes it for the $k-$smallest singular values when the matrix is not invertible. Further, the computation of the change-of-basis matrix $T$ has not been a computational bottle neck in our applications, and our focus is rather to get a well behaved (in sense of retaining sparsity) orthonormal basis. However, we make no claim that our algorithm is optimal, and we rather suspect that an expert on these types of methods could probably improve of algorithm to get a better $T$. This is certainly an interesting topic for further studies.
>
> >The specifics of the covariance-based method are not explained nor is it disclosed which particular algorithm was used. Moreover, no comparison to sparse inverse covariance estimation methods has been performed. In that regard, the proposed GMRF methods are compared solely to other exact methods. In some cases, an approximation might exhibit sufficient performance, while being more efficient.
>
> Regarding covariance-based method we have used the standard Cholesky decomposition method, we will add reference to this.
> Regarding comparison to sparse inverse covariance estimation method, there is to our knowledge no alternative when one adds hard constraints. We agree that  approximate methods often exhibit sufficient performance. The issue is that it would be a completely new paper to design an approximate method that works with hard constraints. We mention in the discussion that an interesting topic for future research would be to combine the proposed methods with approximate Krylov subspace methods for GMRFs, but it is currently not clear to us how exactly this would be done.
>
> In our constrained Gaussian processes example we use an approximation of the gradient operator  (that convergence to the true gradient operator as the mesh is refined) and thus are able to generate a sparse approximation, so that we can use our proposed method. This is one type of efficient sparse approximation which to our knowledge is novel.
>
> >A more thorough experimental evaluation is needed. The main limitations in this regard are summarized as follows: There is an analysis on how the number of observations or basis functions affect the RMSE and the sampling/prediction time. However, there is no analysis on how any of the proposed GMRF variants performs under different levels of sparsity. Has such an experiment been conducted? If so, I would encourage the authors to elaborate on their observations.
>
> We have added more values of $\alpha$ in the first simulation study, which changes the sparsity structure. Specifically, a higher value of $\alpha$ gives less sparsity (we used $\alpha=2$ before) and thus worse performance for the GMRF methods. This changes the timings for the proposed method and for the old GMRF method, but the images look similar. We will add them to the appendix of the paper. Since we cannot add images in this response, we here show a table for a fixed value of $k$ for the first simulation study. In the table, we use $k=3000$ and show average times (in seconds) over ten simulations. In the table "GMRF new" is the total time for our method and we also show the total times for the covariance-based method and the old GMRF method as comparisons.
>
> | $\alpha$ | GMRF new | number of non zero elements in Q | covariance based | GMRF old |
> |---------|----------|-------------------------|------------------|----------|
> | 1       | 1.4      | 49600                     | 11.4              | 9.7      |
> | 2       | 1.3      | 128004                    | 11.2              | 13.0      |
> | 3       | 1.7      | 244420                    | 11.1            | 16.0     |
> | 4       | 2.1     | 398060                   | 11.2             | 21.3     |
>
> Note that for higher values of $\alpha$, the computation time for the proposed method increases, but even for $\alpha=4$ it is still much faster than the covariance-based method (which is not the case for the old GMRF method).
>
> >Too many assumptions are made in the synthetic data generation process. I am wondering if the experiments with GMRF with hard constraints have been repeated for other values of,$(\kappa^2,\phi,\alpha)$,  (other than the [0.5, 1, 2] setting)?
>
> The values of $\kappa^2$ and $\phi$ do not change the sparsity structure of the precision matrix and thus will not affect the results. We therefore just took some reasonable values for the parameters for the first example. We will add a sentence in the article to more strongly emphasis this. The only parameter that affects the results is $\alpha$, and we nave now redone the simulations with different values of $\alpha$ (see the previous response).
>
> >Moreover, in the experiments with the constrained Gaussian processes,  and $\sigma_2$ are simply fixed to $0.01$ and $10^{-4}$, respectively, with no explanation.
>
> We just choose the parameters that where set in [1]. Again, these will not affect the sparsity structure and the specific values are therefore not of any importance for the simulation study.
>
> [1] Carl Jidling et al. “Linearly constrained Gaussian processes”. In: Advances in Neural Information Processing Systems 30. Ed. by I. Guyon et al. Curran Associates, Inc., 2017, pp. 1215–1224

---

### Official Review · Reviewer_2z31 · 2021-07-18

**Rating:** 6
**Confidence:** 3

**Summary:**

This paper proposes a basis transformation method to speed up inference in Gauss--Markov random fields under certain sparse constraints.



**Ethical Concerns:**

no concerns.

**Limitations And Societal Impact:**

yes

**Main Review:**


- Typo? (Line 74)

I believe the first minus sign should be a plus in the conditional mean expression.

- sum-to-zero constraints a violation of assumptions? (Line 191)

It would seem like a sum-to-zero constraint would violate the assumptions on the constraint matrix from Sec 3.1?

- y-axis in Fig 1.

It might be preferable to use a logarithmic scaling of the y-axis for readability.

- Computational time decreasing with number of observations (Line 269)

This would require further explanation.



**Time Spent Reviewing:**

1.5

---

> ### Author Response · Authors · 2021-08-10
> **Reply to  Reviewer 2z31**
>
> Thank you for your comments, please find our response to each comment below.
>
> > Typo? (Line 74)
>
> No, there should not be a plus sign since we are working with precision matrices instead of covariance matrices.
>
> >  sum-to-zero constraints a violation of assumptions? (Line 191)
>
> Yes, in the sense that a sum-to-zero constraint is not a sparse constraint. We will adjust this to say local sum-to-one constraints, which for example can be used when modelling the time varying support (in percentage of voters)  that political parties have.
>
> > y-axis in Fig 1.
>
> Yes, you might be right. We will try this for the revision.
>
> > Computational time decreasing with number of observations (Line 269)
>
> This is because $\mathcal{U}$ decreases and thus the matrix ${\bf Q}^*_{\mathcal{U},\mathcal{U}}$ decreases in size as the number of observations  $\mathcal{U}^c$ decreases. The same happens for sampling, when we need to sample fewer and fewer random variables. We will make a note about this when discussing the results.

---

### Official Review · Reviewer_Ws93 · 2021-07-23

**Rating:** 6
**Confidence:** 4

**Summary:**

The paper proposes a method for evaluating the likelihood and for sampling from the posterior of Gaussian Markov random field $X$ given some linear constraints $X | AX = b$ where $A$ is a sparse matrix. The method is based on the SVD of $A$, which may be computed for a small cost if one can identify rows of $A$ that don't share columns with non-zero elements.

The method is illustrated on two toy examples, where it is shown that it provides massive computational gains. It is however unclear if the competitors considered in the benchmark are the actual alternatives a practitioner would consider.

-----------------------
The author's rebuttal makes me confident that the shortcomings I've identified have been understood by the authors and that they are in a good position to address them at the camera ready stage. I'm thus upgrading my score from 5 to 6.

**Limitations And Societal Impact:**

No specific comments

**Main Review:**

**Writing**

I found the paper to be globally well written and the exposition easy to follow. One shortcoming in my view is that the method could be better motivated (I had to look at the references to convince myself that the problem tackled by the paper had a practical significance). In its current form the paper only specifies that "hard constraints have a cost that scales cubically with the number of constraints" but since this is the core issue that is addressed in this work a little hand holding would be appreciated. In my opinion providing more explanations about why the issue arises and a better positioning of the paper with respect to the prior work (ie how this is currently dealt with) would significantly improve the paper quality.

**Methodology**

The proposed methodology is sound but there is a chance that the assumptions that are added on the sparsity patterns of $A$ when introducing the method could be used in the first place to cut down the complexity of the sampling and of the likelihood evaluation. I believe this could for example be done with the sparse inverse subset method as described in [1].

**Significance**

The feeling that this contribution is of interest to the community would be much stronger if the authors were to give a better description of how the problem they address is currently tackled (or unsolved) and what are the shortcomings. At the moment it seems that they mainly compare a naive baseline which results in impressive results but does not seem to be a fair comparison. For example, my understanding is that using observations as nodes for the triangulation would be a perfectly valid way to tackle the problem in Experiment 6.1.

**Experiments**

Three remarks:
 - Since the proposed method loses its interest when $A$ cannot be split into "independent rows", it would be interesting to increase the number of constraints up to the point where this becomes the case.
 - Could an explanation of why the evaluation time decreases with the number of constraints be included in the result analysis?
 - Is the computation of $T$ included in the timings that are reported?


[1] Durrande, Nicolas, et al. "Banded matrix operators for Gaussian Markov models in the automatic differentiation era." The 22nd International Conference on Artificial Intelligence and Statistics. PMLR, 2019.


**Time Spent Reviewing:**

4

---

> ### Author Response · Authors · 2021-08-10
> **Response to Reviewer Ws93**
>
> Thank you for your comments, please find our responses below.
>
> >One shortcoming in my view is that the method could be better motivated (I had to look at the references to convince myself that the problem tackled by the paper had a practical significance). In its current form the paper only specifies that "hard constraints have a cost that scales cubically with the number of constraints" but since this is the core issue that is addressed in this work a little hand holding would be appreciated. In my opinion providing more explanations about why the issue arises and a better positioning of the paper with respect to the prior work (ie how this is currently dealt with) would significantly improve the paper quality.
>
> GMRFs are popular models for reducing the computational cost of GPs. Because of this, it is of interest to be able to use GMRFs also for constrained GPs, which currently are very popular. For constrained GPs, one often has a large number of constraints, and this is precisely the issue with the existing methods for constrained GMRFs: Their computation time scales cubically in the number of constraints, which removes the computational benefits from the GMRFs. With the introduction of our methods, we believe that GMRFs can become a standard alternative also for constrained GPs. We will add some more details about this in the introduction, as well as some more concrete applications, to further motivate the methods. The positioning of the paper with respect to prior work is dealt with in Section 2, where we introduce the existing methods for constrained GMRFs.
>
> >The proposed methodology is sound but there is a chance that the assumptions that are added on the sparsity patterns of  when introducing the method could be used in the first place to cut down the complexity of the sampling and of the likelihood evaluation. I believe this could for example be done with the sparse inverse subset method as described in [1].
> [1] Durrande, Nicolas, et al. "Banded matrix operators for Gaussian Markov models in the automatic differentiation era." The 22nd International Conference on Artificial Intelligence and Statistics. PMLR, 2019.
>
> In general it is true that one can (and should) use the sparsity patterns, as described [1] and in most other works that deal with GMRFs. However as soon as one adds a large number of hard constraints there has previously not been any methods available to efficiently utilize the sparsity of the constrained model. This is the main goal of the article, to introduce a method that can retain the usage of the sparsity structure of the model (for certain sparse constraints), so that one can use methods like the one described in [1]. We will add further sentences in the introduction that further clarifies this.
>
> >The feeling that this contribution is of interest to the community would be much stronger if the authors were to give a better description of how the problem they address is currently tackled (or unsolved) and what are the shortcomings. At the moment it seems that they mainly compare a naive baseline which results in impressive results but does not seem to be a fair comparison.
>
> The issue is that the only methods we have found for handling hard constraints are those that we present in the article. We will more clearly write in the introduction that this problem has been unsolved until now.
>
> >For example, my understanding is that using observations as nodes for the triangulation would be a perfectly valid way to tackle the problem in Experiment 6.1.
>
> It is true that it is possible in this example to change the mesh so that one has observations only at nodes. However, it is (in general) not a good method as it either requires using very poor meshes (from a numerical stability point of view) or requires a huge amount of mesh nodes, which would increase the computational cost. Further, it is easy to create examples where this is no longer possible, like when one has areal observations or models involving sums of independent Markov random fields. We choose to illustrate the method using this example as it is the simplest, but we will make a remark to clarify this.
>
> >Since the proposed method loses its interest when  cannot be split into "independent rows", it would be interesting to increase the number of constraints up to the point where this becomes the case.
>
>
> In the first experiment we have now run up to $k=5500$ observations. When the number of observations becomes large the constraints may no longer be "local" since many constraints will be interacting. This can be seen in the table below, where the largest set of connected observations is increasing rapidly when $k$ goes above $5000$. The computation time for creating $T$ will roughly scale cubicly in the size of the largest set of connected observations, and the time it takes to evaluate the likelihood will also increase since the sparsity decreases. In the preliminary table below (which will be replaced by an updated image in the revision), we show average times (in seconds) over ten simulations. In the table "GMRF new" is the total time for our method and  we also show the total times for the covariance-based method and the old GMRF method as comparisons.
>
> | $k$  | GMRF new | time to build $T$ | lagest set of connected observations | covariance based | GMRF old |
> |------|----------|-------------------|--------------------------------------|------------------|----------|
> | 1000 | 1.8      | 0.01              | 6.7                                  | 0.6              | 0.8      |
> | 2500 | 1.3      | 0.02              | 24.2                                 | 6.8              | 6.0      |
> | 4000 | 1.0      | 0.10              | 90.7                                 | 25.4             | 19.5     |
> | 4500 | 1.2      | 0.23              | 183.8                                | 35.8             | 27.2     |
> | 5000 | 2.4      | 0.96              | 327.0                                | 48.6             | 35.6     |
> | 5500 | 38.8     | 28.64             | 1023.1                               | 64.6             | 46.1     |
>
> It should be noted that $T$ only needs to be built once. So even though it takes a lot of time to build $T$ when the largest set of connected observations becomes large, the method will still be efficient in numerical optimization of the likelihood where several likelihood evaluations are needed.
>
> >Could an explanation of why the evaluation time decreases with the number of constraints be included in the result analysis?
>
> The reason is that $\mathcal{U}$ decreases and thus the matrix ${\bf Q}^*_{\mathcal{U},\mathcal{U}}$ decreases in size as the number of observations  $\mathcal{U}^c$ decreases.The same happens when sampling, since we need to sample fewer and fewer random variables. We will mention this in connection with the analysis of the results.
>
> >Is the computation of $T$ included in the timings that are reported?
>
> Yes, for all timings. Note that the computation of $T$ only needs to be done once, so in reality the proposed method will be even faster compared to the other methods for both likelhood evaluations and for sampling.

---

> > ### Comment · Reviewer_Ws93 · 2021-08-31
> > **post rebuttal discussion**
> >
> > I'd like to thank the authors for their detailed answer. It gives me confidence that they can address the comments I've raised and I have thus upgraded my score to 6.

---

### Decision · Program_Chairs · 2021-09-27

**Decision:**

Accept (Poster)

**Comment:**

Following reviewers discussion, I recommend accepting the paper under the condition that the authors will explain that its main contributions are applications for the projective clustering problem  in probability/statistics, and cite the relevant existing solutions from different fields (coresets in computational geometry, subspace clustering in DB, dictionary learning in signal processing). It should also be clarified that the suggested solution is given as a simple and possibly inefficient solver of projective clustering for the special case where no noise exists (the points lie on the subspace).
In this case, the paper can serve as an interesting bridge between the CS, ML and statistics community.